# WIN RATE IS ALL THAT CAN MATTER FROM PREFERENCE DATA ALONE

## ABSTRACT

The surging interest in learning from preference data has resulted in an elaborate landscape of methods and evaluations. This work offers a framework to simplify this landscape, starting from the underlying sampling distribution for preference data. First, we show that the only evaluation of a generative model that is grounded in the preference data sampling distribution is win rate. Given that win rate is all that can matter from preference data alone, we relate common preference learning algorithms to direct win rate optimization (DWRO). We outline the theoretical benefits of RLHF as a variant of DWRO; explain why checkpointing is difficult with DPO as a non-DWRO objective; and characterize the limits of SFT on preferred samples with regard to the extent of win rate improvement possible. Furthermore, we provide closed-form expressions for the expected win rate improvement of the above objectives, formalizing the role of a model's starting point in the win rate improvement possible. Finally, we conduct an empirical analysis of existing methods and alternative DWRO objectives which suggests that optimization improvements are likely key to advancing preference learning.

## 1 INTRODUCTION

Learning from preference data, often referred to as human feedback, has emerged as a key step in training large language models, particularly given the success of reinforcement learning from human feedback (RLHF) (Christiano et al., 2017) on state-of-the-art and high-profile language models such as GPT-4 (OpenAI, 2024). The goal of learning from preference data is to finetune powerful base language models to output generations more in line with human preferences (Stiennon et al., 2020; Ouyang et al., 2022), motivated by the fact that pretraining on internet-scale data has enabled large language models to exhibit fluent generations of text (Minaee et al., 2024) but not necessarily responses aligned with what humans prefer to see.

In recent years, the space of algorithms and evaluations for preference learning has grown substantially (Kaufmann et al., 2023; Jiang et al., 2024), resulting in a complex landscape of methods and analyses. Consequently, it can be difficult to pinpoint a clear, overarching framework to compare disparate works and to guide progress. How can we understand preference learning from the ground up?

A common approach to describe preference learning is to focus on the historical development of methods in the space. Such a description typically starts with reinforcement learning from human feedback (RLHF) (Christiano et al., 2017), which involves learning a reward model from preference data and optimizing the language model policy to maximize the learned rewards. Follow-up works can generally be described as efforts to improve upon RLHF, among which direct preference optimization (DPO) (Rafailov et al., 2024) was developed to estimate the same solution but via a single step of direct finetuning rather than a two-step procedure of RLHF. In other words, the landscape of preference learning can often be described by how methods relate to each other, typically connecting back to RLHF as a focal point. In this work, in contrast, we ask: is there another way to understand preference learning that does not center around the earliest or most popular methods?

We address this question by developing a framework for understanding the endeavor of learning from preferences starting from the sampling distribution implied by pairwise preference data. We first show that the only evaluation of a generative model rooted in the preference data sampling distribution itself is win rate; any other notions of good are a function of outside assumptions (Section 3). From this insight, we introduce a win rate-centric framework for understanding preference learning

(Section 4, Section 5). Given win rate is the only relevant evaluation without additional assumptions, we relate common preference learning algorithms (RLHF, DPO, SFT) to directly optimizing for win rate, outlining benefits of RLHF for being a variant of direct win rate optimization (DWRO) and limitations of DPO and SFT for not. Namely, as a variant of DWRO, RLHF confers benefits from (1) optimizing train loss corresponds to optimizing for the test evaluation we care about (up to noise and overfitting), which is not true of DPO and related variants; to (2) its solution can achieve the maximum win rate possible over a competitor as regularization strength goes to zero, unlike SFT. These insights are relevant not just to the three methods discussed, but also for other methods in the same family (e.g., DWRO variants other than RLHF, Direct Alignment Algorithms other than DPO). We then conduct an empirical comparison of different methods which offers complementary insights to the theoretical analysis (Section 6). Namely, we show that despite their theoretical benefits, direct win rate optimization methods underperform relative to expectations due to difficulties in optimization. We conclude by discussing the takeaways from our combined theoretical and empirical analysis (Section 7). These include explanations to ground specific strategies or current practices in preference learning, as well as guidance for future research.

Our contributions can be summarized as follows:

1. We prove that the only evaluation of a generative model grounded in the preference data distribution is win rate. This result justifies using win rate as a singular focal point to understand the landscape of preference learning.

2. We present a win rate-centric framework to understand preference learning. From our combined theoretical and empirical analysis under this framework, we:

   (a) present theoretical benefits of methods which directly optimize for win rate (e.g., RLHF) and limitations of methods which do not (e.g., DPO, SFT);

   (b) experimentally demonstrate the challenges of optimizing for win rate and the central role of optimization success for the performance of different methods;

   (c) discuss implications for the current practice of choosing methods;

   (d) bring attention to connections between preference learning and probabilistic inference to help inform future work.

## 2 RELATED WORK

Our work is most closely related to previous work in win rate evaluation and optimization as well as analysis of RLHF and preference learning objectives.

**Win rate evaluation and optimization.** Win rate is already a central evaluation in preference learning (Li et al., 2023; Zheng et al., 2024); however, our work goes further to underscore that it is the only evaluation grounded in the sampling distribution itself, thus motivating its use as the central object to understand the rest of the preference learning landscape, including analytically. Several works have proposed methods that perform some form of win rate optimization (Munos et al., 2023; Swamy et al., 2024; Rosset et al., 2024). Our work emphasizes that win rate optimization is central goal of preference learning overall, analyzing a spectrum of preference learning methods through this lens and pinpointing existing bottlenecks to address to better realize this goal.

**Analyzing RLHF, DPO, and other preference learning methods.** Our work is related to work that seeks to better understand RLHF, DPO, and other existing methods in preference learning (e.g., best-of-n). Examples include benchmarking generalization and diversity (Kirk et al., 2024), comparing on- vs off-policy approaches (Tajwar et al., 2024), investigating length bias (Singhal et al., 2024), and disentangling design choices empirically (Ivison et al., 2024). For RLHF in particular, existing works consider the complexity of proximal policy optimization (Ahmadian et al., 2024), vanishing gradients (Razin et al., 2024b), reward model overoptimization (Zhu et al., 2024), or limitations of the Bradley-Terry assumption to relate preferences to rewards (Wang et al., 2024a; Azar et al., 2023; Munos et al., 2023). For DPO, there exists not only large space of alternative direct alignment algorithms (e.g., (Zhao et al., 2023; Azar et al., 2023; Xu et al., 2024a; Huang et al., 2024; Pal et al., 2024; Xu et al., 2024b)) but also methods which analyze its limited ability to flip rankings (Chen et al., 2024) or the decrease in chosen and rejected log probabilities (Razin et al., 2024a). Gui et al. (2024) also analyze the target distributions of different preference learning algorithms but focuses on the optimality of best-of-n while we showcase the limits of SFT as well. Our work is most closely

related to Azar et al. (2023): their $\Psi$-Preference Optimization objective is direct win rate optimization with a KL regularizer, and they are the first to show that RLHF falls within the family of DWRO-KL objectives. However, while they use the analysis to motivate their proposed IPO as an alternative to DPO, we use DWRO to contrast RLHF from direct alignment algorithms like DPO and IPO, highlighting the theoretical benefits of the former and limitations of the latter. Moreover, rather than focus a direct alignment algorithm variant of a particular instantiation within the family of objectives, we run experiments on a wide range of DWRO objectives and demonstrate that design choices within such objectives matter less to performance than the optimization success of a given training run.

# 3 PREFERENCE LEARNING SETUP

Here, we outline the setup for learning from pairwise preferences. We first describe the underlying sampling distribution of the data used in preference learning (Section 3.1). Then we show why based on the sampling distribution alone, the only evaluation which respects the underlying preference environment and sampling prevalences is win rate (Section 3.2).

## 3.1 THE SAMPLING DISTRIBUTION

The goal of preference learning is to learn a generative model that performs well in a given context. However, whereas typical maximum likelihood training employs samples from the distribution of interest, the setup of preference learning does not: only samples from generation competitors and their relative preference under the query-preference environment are available.

The sampling distribution for preferences involves input $\mathbf{x}$, candidate outputs $\mathbf{y}_0$ and $\mathbf{y}_1$, and a label $\ell \in \{0, 1\}$ denoting which of $\mathbf{y}_0$ or $\mathbf{y}_1$ is preferred. Let $\ell = 1$ denote that $\mathbf{y}_1$ is preferred, and $\ell = 0$ denote that $\mathbf{y}_0$ is preferred. Then, the overall sampling distribution can be defined as follows:

**Definition 1.** *A **sampling distribution for (pairwise) preference learning** is a distribution over input $\boldsymbol{x} \in \mathcal{X}$, candidate outputs $\mathbf{y}_0, \mathbf{y}_1 \in \mathcal{Y}$, and preference label $\ell \in \{0, 1\}$ defined by:*

*1. Query distribution: $p(\boldsymbol{x})$*

*2. Generation competitor 0: $p(\mathbf{y}_0 \,|\, \boldsymbol{x})$*

*3. Generation competitor 1: $p(\mathbf{y}_1 \,|\, \boldsymbol{x})$*

*4. Preference classifier: $p(\ell \,|\, \boldsymbol{x}, \mathbf{y}_0, \mathbf{y}_1)$.*

*1, 2, and 3 are user-specified distributions; 1 denotes the inputs of interest, and 2 and 3 are the candidate competitors one chooses to evaluate. 4 is only distribution that cannot be directly specified, rather it is defined by the environment in which the user choose to collect the preferences.*

Generation competitor 0 and 1 can be the same distribution and often are in existing open-source preference datasets (Lambert et al., 2023; Ethayarajh et al., 2022).

## 3.2 WIN RATE IS THE ONLY EVALUATION THAT CAN MATTER

The goal of preference learning is to learn some generative model $p^*(\mathbf{y} \,|\, \mathbf{x})$ that performs well under the preference environment for a given query distribution (we refer to this as the *query-preference environment*). Learning such a generative model requires a definition of what is good. Consider an evaluation function $\phi$ which maps a generative model $p(\mathbf{y} \,|\, \mathbf{x})$, query-preference environment $\mathcal{E} = (p(\mathbf{x}), p(\ell = 1 \,|\, \mathbf{x}, \mathbf{y}_0, \mathbf{y}_1))$, and *anchor distribution* $p(\mathbf{y}_0 \,|\, \mathbf{x})$ to a scalar: $\phi_{p(\mathbf{y}_0 \,|\, \mathbf{x})}(p(\mathbf{y} \,|\, \mathbf{x}), \mathcal{E}) \in \mathbb{R}$. We will optionally write $\phi_{p(\mathbf{y}_0 \,|\, \mathbf{x})}$ as $\phi$ when the anchor is clear from the context.

Intuitively, an evaluation function $\phi$ should respect properties of the preference sampling distribution. We formalize these properties for $\phi$ to respect in the definition below.

**Definition 2.** *Any evaluation function $\phi$ is **grounded** in a given preference distribution if*

*1. (base case is a function of preference classification): given query environment $p(\boldsymbol{x}) = \mathbb{1}\,[\boldsymbol{x} = \boldsymbol{x}']$, generative model $p(\mathbf{y} \,|\, \boldsymbol{x}) = \mathbb{1}\,[\mathbf{y} = \mathbf{y}']$, anchor distribution $p(\mathbf{y}_0 \,|\, \boldsymbol{x}) = \mathbb{1}\,[\mathbf{y}_0 = \mathbf{y}_0']$, and strictly monotonic increasing function $h$:*
$$\phi(p(\mathbf{y} \,|\, \boldsymbol{x}), \mathcal{E}) = h \cdot p(\ell = 1 \,|\, \boldsymbol{x}', \mathbf{y}_0', \mathbf{y}'); \textit{ and}$$

2. *(respects prevalences in generator, anchor, and query): for $a, b \geq 0$ and $a + b = 1$:*

   *for generator $p(\boldsymbol{y} \mid \boldsymbol{x}) = a p_1(\boldsymbol{y} \mid \boldsymbol{x}) + b p_2(\boldsymbol{y} \mid \boldsymbol{x})$:*

$$\phi(p(\boldsymbol{y} \mid \boldsymbol{x}), \mathcal{E}) = a\phi(p_1(\boldsymbol{y} \mid \boldsymbol{x}), \mathcal{E}) + b\phi(p_2(\boldsymbol{y} \mid \boldsymbol{x}), \mathcal{E}); \tag{1}$$

   *for query distribution $p(\boldsymbol{x}) = a p_1(\boldsymbol{x}) + b p_2(\boldsymbol{x})$, letting $\mathcal{E}_i = (p_i(\boldsymbol{x}), p(\ell = 1 \mid \boldsymbol{x}, \boldsymbol{y}_0, \boldsymbol{y}_1))$:*

$$\phi(p(\boldsymbol{y} \mid \boldsymbol{x}), \mathcal{E}) = a\phi(p(\boldsymbol{y} \mid \boldsymbol{x}), \mathcal{E}_1) + b\phi(p(\boldsymbol{y} \mid \boldsymbol{x}), \mathcal{E}_2); \text{ and} \tag{2}$$

   *for anchor distribution $p(\boldsymbol{y}_0 \mid \boldsymbol{x}) = a p_1(\boldsymbol{y}_0 \mid \boldsymbol{x}) + b p_2(\boldsymbol{y}_0 \mid \boldsymbol{x})$ :*

$$\phi_{p(\boldsymbol{y}_0 \mid \boldsymbol{x})}(p(\boldsymbol{y} \mid \boldsymbol{x}), \mathcal{E}) = a\phi_{p_1(\boldsymbol{y}_0 \mid \boldsymbol{x})}(p(\boldsymbol{y} \mid \boldsymbol{x}), \mathcal{E}) + b\phi_{p_2(\boldsymbol{y}_0 \mid \boldsymbol{x})}(p(\boldsymbol{y} \mid \boldsymbol{x}), \mathcal{E}). \tag{3}$$

Condition 1 ensures that the evaluation can be reduced to a function of the preference classifier in the limiting case where everything else is a point mass, and condition 2 ensures that the contribution of the preference classification $p(\ell = 1 \mid \mathbf{x}, \mathbf{y}_0, \mathbf{y}_1)$ for a given $(\mathbf{x}, \mathbf{y}_0, \mathbf{y}_1)$ gets weighted appropriately by its prevalence in the query environment, anchor distribution, and generative model, respectively. The only evaluation that satisfies Definition 2 is some form of win rate:

**Proposition 1.** *$\phi$ is grounded, as defined in Definition 2, if and only if*

$$\phi_{p(\mathbf{y}_0 \mid \mathbf{x})}(p(\boldsymbol{y} \mid \boldsymbol{x}), \mathcal{E}) = \mathbb{E}_{p(\boldsymbol{x})}\mathbb{E}_{p(\boldsymbol{y} \mid \boldsymbol{x})}\mathbb{E}_{p(\boldsymbol{y}_0 \mid \boldsymbol{x})}[h \cdot p(\ell = 1 \mid \boldsymbol{x}, \boldsymbol{y}_0, \boldsymbol{y})] \tag{4}$$

*for some choice of $p(\mathbf{y}_0 \mid \mathbf{x})$ and strictly monotonic increasing function h.*

See Appendix A for proof. Equation (4) is the win rate of the generator $p(\mathbf{y} \mid \mathbf{x})$ against anchor distribution $p(\mathbf{y}_0 \mid \mathbf{x})$ under the query-preference environment $\mathcal{E}$ for some choice of order-preserving transformation function $h$. When $h$ is the identity, we have vanilla win rate under the preference environment; we will refer to this variant as win rate and equivalently write $\text{Win Rate}_{p(\mathbf{y}_0 \mid \mathbf{x})}[p(\mathbf{y} \mid \mathbf{x})]$. Often in automated win rate evaluations, it is estimated via samples from the preference classifier, i.e., $\mathbb{E}_{p(\mathbf{x})}\mathbb{E}_{p(\mathbf{y} \mid \mathbf{x})}\mathbb{E}_{p(\mathbf{y}_0 \mid \mathbf{x})}\ell$, where $\ell \sim p(\ell \mid \mathbf{x}, \mathbf{y}_0, \mathbf{y})$ (Li et al., 2023). When $h$ is any other strictly monotonic increasing function, we have an $h$-variant of win rate.

Note that placing non-identity functions in any other position in Equation (4), i.e., $\mathbb{E}_{p(\mathbf{x})} f \cdot \mathbb{E}_{p(\mathbf{y} \mid \mathbf{x})} g \cdot \mathbb{E}_{p(\mathbf{y}_0 \mid \mathbf{x})}[h \cdot p(\ell = 1 \mid \mathbf{x}, \mathbf{y}_0, \mathbf{y})]$, breaks the second property of respecting prevalences in Definition 2 by giving outsized priority to certain elements in the generator and query distribution.

Given the insight that the only evaluation grounded in preference data alone is some form of win rate with respect to some anchor distribution, we next analyze the relationship between common preference learning algorithms and win rate.

## 4 PREFERENCE LEARNING THROUGH THE LENS OF WIN RATE OPTIMIZATION

In the next two sections, we analyze existing preference learning algorithms based on how they relate to directly optimizing for win rate. First, we introduce Direct Win Rate Optimization (DWRO) as a focal point for understanding the preference learning landscape (Section 4.1). Then, we analyze how existing preference learning algorithms relate to it, providing insights into the benefits and limitations of different methods.

### 4.1 DIRECT WIN RATE OPTIMIZATION

The only grounded evaluation developed in Equation (4) immediately provides an objective to optimize:

$$\max_{\theta} \text{Win Rate}_{p(\mathbf{y}_0 \mid \mathbf{x})}[p_\theta(\mathbf{y}_1 \mid \mathbf{x})] = \max_{\theta} \mathbb{E}_{p(\mathbf{x})}\mathbb{E}_{p_\theta(\mathbf{y}_1 \mid \mathbf{x})}\mathbb{E}_{p(\mathbf{y}_0 \mid \mathbf{x})}[h \cdot p(\ell = 1 \mid \mathbf{x}, \mathbf{y}_0, \mathbf{y}_1)]. \tag{5}$$

This objective maximizes a function $h$ of the preference probability in expectation over an anchor distribution $p(\mathbf{y}_0 \mid \mathbf{x})$. We refer to Equation (5) as the family of **D**irect **W**in **R**ate **O**ptimization, or DWRO, objectives. The solution to any objective in this family is the generator that maximizes the $h$-win rate over a given anchor distribution.

While in general different choice of $h$ can yield different solutions, under the Bradley-Terry assumption, all DWRO-$h$ objectives maximize win rate and all $h$-variants of it.

**Proposition 2.** *(informal) Under the Bradley-Terry assumption, all DWRO objectives with strictly monotonic increasing $h$ share the same optimal solution.*

See Appendix B for proof. This result states that when pairwise preferences can be mapped to a global ranking of sequences by preference, then perfectly optimizing any DWRO objective yields a solution that is a perfect optimum of any other DWRO objective.

**Regularization.** Preference classifiers train on a fixed collection of data samples, which means there will be inevitable estimation error in $p(\ell = 1 \,|\, \mathbf{x}, \mathbf{y}_0, \mathbf{y}_1)$. Any DWRO objective would optimize and overfit to these errors. Preventing this kind of overfitting requires regularization, which can be accomplished with a divergence penalty with regularization parameter $\beta$:

$$-\mathcal{L}_{\text{DWRO-reg}}(\theta) = \max_{\theta} \mathbb{E}_{p(\mathbf{x})}\big[\mathbb{E}_{p_\theta(\mathbf{y}_1 \,|\, \mathbf{x})}\mathbb{E}_{p(\mathbf{y}_0 \,|\, \mathbf{x})}[h \cdot p(\ell = 1 \,|\, \mathbf{x}, \mathbf{y}_0, \mathbf{y}_1)] - \beta D(p_\theta, p_{\text{ref}})\big]. \quad (6)$$

Options for $D$ include sequence-level reverse KL divergence or chi-sq divergence, considered in Azar et al. (2023) and Huang et al. (2024) respectively, and a sum of token-level divergences.

**Benefits of DWRO objectives.** There are two benefits to DWRO objectives:

1. A correspondence between the training objective and grounded evaluations
2. The ability to maximize the grounded evaluation of $h$-win rate up to any regularization

The first benefit follows directly from the fact that DWRO objectives optimize for the evaluation of interest, so improvements in train loss correspond to improvements in the test evaluation, up to overfitting and noise. The second benefit also follows from optimizing the evaluation of interest. Directly optimizing for win rate means no limits on how much the evaluation can be improved. In contrast, as we see in the section Section 5, these properties need not be true for other losses that are not DWRO objectives.

**Optimization of DWRO objectives.** When the divergence $D$ is tractable, DWRO can be optimized by differentiating the divergence and using score function gradients (Mohamed et al., 2020a). When the divergence is intractable but the entire objective can be written as the form $\mathbb{E}_{p(\mathbf{x})}\mathbb{E}_{p_\theta(\mathbf{y} \,|\, \mathbf{x})}\phi(\mathbf{x}, \mathbf{y})$, score function gradients or or another policy gradient algorithm (Weng, 2018) can still be used for optimization.[1]

Optimization algorithms for DWRO of this type require estimating $\mathbb{E}_{p(\mathbf{y}_0 \,|\, \mathbf{x})}[h \cdot p(\ell = 1 \,|\, \mathbf{x}, \mathbf{y}_0, \mathbf{y}_1)]$, which can be done by first learning the preference distribution $p(\ell = 1 \,|\, \mathbf{x}, \mathbf{y}_0, \mathbf{y}_1)$ from data and then estimating the expectation $\mathbb{E}_{p(\mathbf{y}_0 \,|\, \mathbf{x})}[h \cdot p(\ell = 1 \,|\, \mathbf{x}, \mathbf{y}_0, \mathbf{y}_1)]$ with a sample or samples from $p(\mathbf{y}_0 \,|\, \mathbf{x})$ during policy optimization. Alternatively, it can be possible to learn a model that estimates this expectation directly in the first step, removing the need to additionally sample from $p(\mathbf{y}_0 \,|\, \mathbf{x})$ during the second policy optimization step.

Optimizing DWRO with reverse KL regularization is equivalent to minimizing the reverse KL divergence between the model and the target distribution $p^*_{\text{DWRO-KL}}(\mathbf{y} \,|\, \mathbf{x}) \propto p_{\text{ref}}(\mathbf{y} \,|\, \mathbf{x}) \exp(\frac{1}{\beta}\mathbb{E}_{p(\mathbf{y}_0|\mathbf{x})}[h \cdot p(\ell = 1|\mathbf{x}, \mathbf{y}_0, \mathbf{y})])$; see Azar et al. (2023) or Appendix C.0.1 for derivation. As such, the reverse-KL regularized DWRO objective is analogous to that of variational inference (Blei et al., 2017). In particular, it is a form of black-box variational inference (Ranganath et al., 2014).

**Benefits of RLHF as a DWRO Objective.** KL-constrained RLHF utilizes the Bradley Terry (BT) assumption (Bradley & Terry, 1952) to define a reward model $r(\mathbf{x}, \mathbf{y})$ such that $p(\ell = 1 \,|\, \mathbf{x}, \mathbf{y}_0, \mathbf{y}_1) = \sigma[r(\mathbf{x}, \mathbf{y}_1) - r(\mathbf{x}, \mathbf{y}_0)]$, then maximizes this reward with a KL penalty:

$$-\mathcal{L}_{\text{RLHF}}(\theta) = \max_{\theta} \mathbb{E}_{p(\mathbf{x})}\big[\mathbb{E}_{p_\theta(\mathbf{y} \,|\, \mathbf{x})}[r(\mathbf{x}, \mathbf{y})] - \beta \text{KL}(p_\theta(\mathbf{y} \,|\, \mathbf{x}) \,\|\, p_{\text{ref}}(\mathbf{y} \,|\, \mathbf{x}))\big]. \quad (7)$$

As proved in Azar et al. (2023), RLHF is a DWRO-KL objective under the BT assumption, where the transformation function $h$ is the logit function:

$$-\mathcal{L}_{\text{RLHF}}(\theta) = \max_{\theta} \mathbb{E}_{p(\mathbf{x})}\big[\mathbb{E}_{p_\theta(\mathbf{y} \,|\, \mathbf{x})}\mathbb{E}_{p(\mathbf{y}_0 \,|\, \mathbf{x})}[\text{logit}\, p(\ell = 1 \,|\, \mathbf{x}, \mathbf{y}_0, \mathbf{y})] - \beta \text{KL}(p_\theta(\mathbf{y} \,|\, \mathbf{x}) \,\|\, p_{\text{ref}}(\mathbf{y} \,|\, \mathbf{x}))\big].$$

---

[1]For instance, for DWRO-$\chi^2$, $\phi(\mathbf{x}, \mathbf{y}) = \mathbb{E}_{p(\mathbf{y}_0 \,|\, \mathbf{x})}[h \cdot p(\ell = 1 \,|\, \mathbf{x}, \mathbf{y}_0, \mathbf{y}_1)] - \left(\frac{p_{\text{ref}}(\mathbf{y} \,|\, \mathbf{x})}{p_\theta(\mathbf{y} \,|\, \mathbf{x})} - 1\right)^2$; for DWRO-KL, $\phi(\mathbf{x}, \mathbf{y}) = \mathbb{E}_{p(\mathbf{y}_0 \,|\, \mathbf{x})}[h \cdot p(\ell = 1 \,|\, \mathbf{x}, \mathbf{y}_0, \mathbf{y}_1)] - \log p_\theta(\mathbf{y} \,|\, \mathbf{x}) + \log p_{\text{ref}}(\mathbf{y} \,|\, \mathbf{x})$.

This means that RLHF benefits from properties 1 and 2. In other words, under the Bradley-Terry assumption, optimizing the RLHF objective is optimizing for the evaluation we care about. This perspective offers an alternative motivation for RLHF as a central method in preference learning, beyond its status as the first to popularize preference learning. However, RLHF is just one potential method to capture the idea of optimizing directly for the evaluation we care about. Do RLHF's specific choices confer any additional benefits?

**Benefits of RLHF relative to other DWRO objectives.** While optimizing DWRO-KL objectives in general require sampling from the current and initial model to estimate win rate, RLHF's specific choice of BT assumption and $h = \text{logit}$ allows its objective to be computed as a function of the current model output only: optimizing $p_\theta(\mathbf{y} \mid \mathbf{x})$ for $\mathbb{E}_{p(\mathbf{y}_0 \mid \mathbf{x})}[r(\mathbf{x}, \mathbf{y}) - r(\mathbf{x}, \mathbf{y}_0)]$ is equivalent to optimizing for $r(\mathbf{x}, \mathbf{y})$, since $\mathbb{E}_{p(\mathbf{y}_0 \mid \mathbf{x})} r(\mathbf{x}, \mathbf{y}_0)$ is a constant with respect to $p_\theta(\mathbf{y} \mid \mathbf{x})$. In other words, the RLHF variant of DWRO-KL offers a computational advantage that the other variants do not: its score can be computed without sampling any reference model outputs. Dropping $\mathbb{E}_{p(\mathbf{y}_0 \mid \mathbf{x})} r(\mathbf{x}, \mathbf{y}_0)$ from the objective also confers optimization advantages with respect to variance of the gradient estimator.

**Takeaways.** The DWRO properties 1 and 2 confer important benefits for learning from preference data. RLHF enjoys these benefits, as well as computational efficiency and reduced gradient variance relative to other DWRO objectives.

Next we analyze to objectives that are used in preference learning but are *not* direct win rate optimization objectives.

## 5 PREFERENCE ALGORITHMS THAT ARE NOT DWRO OBJECTIVES

### 5.1 DPO SHARES THE SAME TARGET AS RLHF BUT DOES NOT OPTIMIZE WIN RATE

Direct Preference Optimization (DPO) (Rafailov et al., 2024) shares the same target distribution as RLHF but employs a different objective to optimize for that target, namely:

$$-\mathcal{L}_{\text{DPO}}(\theta) = \min_\theta \mathbb{E}_{p(\mathbf{x}, \mathbf{y}_0, \mathbf{y}_1, \ell)} \big[ \text{KL}(p(\ell = 1 \mid \mathbf{x}, \mathbf{y}_0, \mathbf{y}_1) \,\|\, p_\theta(\ell = 1 \mid \mathbf{x}, \mathbf{y}_0, \mathbf{y}_1)) \big], \tag{8}$$

where $p_\theta(\ell = 1 \mid \mathbf{x}, \mathbf{y}_0, \mathbf{y}_1)$ is parametrized to include a language model inside of it:

$$p_\theta(\ell = 1 \mid \mathbf{x}, \mathbf{y}_0, \mathbf{y}_1) = \sigma \Big[ \beta \log \frac{p_\theta(\mathbf{y}_1 \mid \mathbf{x})}{p_{\text{ref}}(\mathbf{y}_1 \mid \mathbf{x})} - \beta \log \frac{p_\theta(\mathbf{y}_0 \mid \mathbf{x})}{p_{\text{ref}}(\mathbf{y}_0 \mid \mathbf{x})} \Big]. \tag{9}$$

DPO targets a DWRO-KL distribution (i.e., the same one as RLHF), but its objective is derived by using the relationship between the target distribution and preference classifier to substitute the former into an objective optimizing the latter. Notably, optimizing the estimation of the preference classifier is not the same as optimizing for win rate. As a result, improvements in DPO loss do not necessarily correspond to improvements in win rate, i.e. property 1 does not hold. This insight offers an explanation for empirical results such as the loss vs. win rate misalignment observed in (Chen et al., 2024), as well as the benefit of early stopping or checkpointing with win rate directly, as is common in current practice (Rafailov et al., 2024; Yuan et al., 2024). This insight also applies to alternative direct alignment algorithms which do not optimize the win rate objective (e.g., Azar et al. (2023); Tang et al. (2024); Huang et al. (2024)).

**Takeaways.** DPO and other direct alignment algorithms do not directly optimize win rate. Thus even if they target the same solution as a DWRO objective (i.e., property 2 holds, true for Rafailov et al. (2024); Azar et al. (2023); Huang et al. (2024)), improvements in loss do not necessarily correspond to improvements in win rate (i.e., property 1 does not hold), resulting in difficulties with model selection without external evaluations outside of loss.

### 5.2 SUPERVISED FINETUNING ON PREFERRED SAMPLES IMPROVES WIN RATE WITH LIMITS

Supervised finetuning (SFT) on preferred samples is often viewed as an initial step or necessary precursor to other preference learning algorithms such as RLHF (Wang et al., 2024b; Razin et al., 2024b) but here, we place it on the equal footing to other preference learning algorithms. First, we show that SFT on preferred samples is not a DWRO objective. Then, we compare its target

distribution to that of DWRO-KL objectives and show that it is limited in its ability to yield sharp distributions that concentrate on elements that win often. Moreover, we characterize the exact win rate expected from SFT with online samples, as well as the insights it offers for self-improvement.

Supervised finetuning on preferred samples (sometimes denoted $\mathbf{y}_w$) seeks to maximize the likelihood of sample $\mathbf{y}_1$ when $\ell = 1$ and $\mathbf{y}_0$ when $\ell = 0$. As $p(\mathbf{y}_w \,|\, \mathbf{x}) = p(\mathbf{y}_1 \,|\, \mathbf{x}, \ell = 1) = p(\mathbf{y}_0 \,|\, \mathbf{x}, \ell = 0)$, the objective can be written as follows:

$$-\mathcal{L}_{\text{SFT}}(\theta) = \min_\theta \mathbb{E}_{p(\mathbf{x})} \text{KL}(p(\mathbf{y}_1 \,|\, \mathbf{x}, \ell = 1) || p_\theta(\mathbf{y}_1 \,|\, \mathbf{x}, \ell = 1)). \tag{10}$$

SFT optimizes for this target distribution through the forward KL objective, which is possible due to the fact that one can obtain samples for the target distribution from the preference sampling distribution directly. The analogous reverse KL objective for this target distribution is as follows (see Appendix C.0.3 for derivation):

$$\max_\theta \mathbb{E}_{p(\mathbf{x})} \left[ \mathbb{E}_{p_\theta(\mathbf{y}_1 \,|\, \mathbf{x})} \log \mathbb{E}_{p(\mathbf{y}_0 \,|\, \mathbf{x})}[p(\ell = 1 \,|\, \mathbf{x}, \mathbf{y}_0, \mathbf{y}_1)] - \text{KL}(p_\theta(\mathbf{y} \,|\, \mathbf{x}) \,\|\, p_{\text{ref}}(\mathbf{y} \,|\, \mathbf{x})) \right]. \tag{11}$$

While Equation (11) looks similar to a DWRO-KL objective, it is not. Concretely, the lack of $\beta$ to scale the KL divergence penalty, as well as the additional $\log$ outside of the innermost expectation, differentiate this objective from DWRO-KL. The result is that neither property 1 or 2 hold.

What is the effect? The target distribution for SFT can be written as:

$$p_{\text{SFT}}^*(\mathbf{y} \,|\, \mathbf{x}) \propto p(\mathbf{y} \,|\, \mathbf{x}) \mathbb{E}_{p(\mathbf{y}_0 \,|\, \mathbf{x})}[p(\ell = 1 \,|\, \mathbf{x}, \mathbf{y}_0, \mathbf{y}_1)], \tag{12}$$

or equivalently

$$p(\mathbf{y}_1 \,|\, \mathbf{x}, \ell = 1) = \frac{p(\mathbf{y}_1, \ell = 1 \,|\, \mathbf{x})}{p(\ell = 1 \,|\, \mathbf{x})} = \frac{\int p(\mathbf{y}_1 \,|\, \mathbf{x}) p(\mathbf{y}_0 \,|\, \mathbf{x}) p(\ell = 1 \,|\, \mathbf{x}, \mathbf{y}_0, \mathbf{y}_1) d\mathbf{y}_0}{\int p(\mathbf{y}_1' \,|\, \mathbf{x}) p(\mathbf{y}_0' \,|\, \mathbf{x}) p(\ell = 1 \,|\, \mathbf{x}, \mathbf{y}_0', \mathbf{y}_1') d\mathbf{y}_0' d\mathbf{y}_1'}. \tag{13}$$

The distribution tilts the original distribution towards sequences with higher average preference probabilities over the anchor, but there are limits to the amount of change Equation (12) can achieve. In fact, we can characterize the exact win rate expected from SFT over the original model:

**Theorem 1** (Win rate improvement of SFT). *Let $p(\mathbf{y}_0 \,|\, \boldsymbol{x})$ be the initial generative model, and $p_{SFT}(\boldsymbol{y} \,|\, \boldsymbol{x})$ be the target distribution of supervised finetuning on preferred samples ($p(\mathbf{y}_1 \,|\, \boldsymbol{x}, \ell = 1)$, $p(\mathbf{y}_0 \,|\, \boldsymbol{x}) = p(\mathbf{y}_1 \,|\, \boldsymbol{x})$). Then,*

$$\textit{Win Rate}_{p(\mathbf{y}_0 \,|\, \boldsymbol{x})}[p_{SFT}(\boldsymbol{y} \,|\, \boldsymbol{x})] = 0.5 + 2\mathbb{E}_{p(\mathbf{x})} \textit{Var}_{p(\mathbf{y}_1 \,|\, \boldsymbol{x})} \left[ \int p(\mathbf{y}_0 \,|\, \boldsymbol{x}) p(\ell = 1 \,|\, \boldsymbol{x}, \mathbf{y}_0, \mathbf{y}_1) d\mathbf{y}_0 \right], \tag{14}$$

*which is less than 1.0 as long as there exist non-deterministic preference probabilities, $p(\ell = 1 \,|\, \boldsymbol{x}, \mathbf{y}_0, \mathbf{y}_1) \in (0, 1)$.*

See Appendix E for proof. The theorem states that variance in the average preference probability of the sequences in the initial model dictates the extent of win rate improvement. Intuitively, one should not expect any possible improvement in win rate if the existing model only outputs sequences which are equally preferred to each other. On the other hand, there is more room for improvement in win rate the more differentially preferred some of the model's sequences are to others. The most improvement in win rate occurs when the variance is high. The win rate would be optimized if the variance were $1/4$, but this is not possible for a random variable between 0 and 1 that with density anywhere besides the endpoints. As a point of reference, if the average preference probabilities are uniform between zero and one for every input, the improvement delta will be $2 \times \frac{1}{12} = \frac{1}{6} \approx 0.167$, meaning a win rate of just $0.667$.

**Takeaways.** SFT on preferred samples is a method that improves win rate but does not optimize it nor targets a DWRO solution. Since neither property 1 or 2 hold, SFT has limits to the amount of win rate improvement possible given preference annotations alone, and the improvement possible is a function of how differentially preferred the starting model's sequences are to each other.

**An aside: Expected win rate improvement of other objectives.** Similar to Theorem 1, closed-form expressions can be derived for the target distributions of DWRO-KL objectives broadly (see Appendix F). While it is difficult to obtain low-variance estimators these expressions (e.g., due to multiple levels of nested expectations, a ratio of expectations), the analytical forms themselves provide insight into what improvements we can expect from preference learning: for instance, for any DWRO-KL with non-zero KL regularization, the expected win rate improvement over the starting model is upper bounded by the maximum average preference probability over the model's sequences:

**Corollary 1.** *For any DWRO-KL objective with non-zero KL regularization, the win rate improvement expected over the starting model is upper bounded by:*

$$Win\ Rate_{p(\mathbf{y}_0 \mid \mathbf{x})}[p^*(\mathbf{y} \mid \mathbf{x})] \leq \mathbb{E}_{p(\mathbf{x})} \left[ \max_{\mathbf{y} \in supp(p(\mathbf{y}_0 \mid \mathbf{x}))} \mathbb{E}_{p(\mathbf{y}_0 \mid \mathbf{x})} p(\ell = 1 \mid \mathbf{x}, \mathbf{y}_0, \mathbf{y}_1) \right].$$

See Appendix G for proof. This result pertains to RLHF, which is a variant of DWRO-KL, as well as DPO, which shares the same target solution. This result, alongside the analytical win rate improvement expressions, emphasizes the central role of the starting model in the win rate improvement possible when learning from preference annotations alone (due to KL regularization). In the case of SFT, the variance of the model's output sequence preference probabilities over the others on average dictates the win rate improvement possible. For DWRO-KL objectives, even though they optimize win rate, the KL regularization forces the resulting target distribution to place mass only over sequences already supported in the original distribution, meaning that the best query-conditional win rate possible is a property of the best sequence's performance relative to the rest.

## 6 INVESTIGATING THE EMPIRICAL IMPACT OF DIFFERENT DESIGN CHOICES

Here, we compare RLHF, DPO, and SFT, in order to complement our above theoretical analysis with an empirical one. In particular, rather than compare the conventional two-stage SFT + RLHF or DPO, we compare all three methods on the same footing by how much they are able to improve win rate over the starting model. Moreover, as RLHF is just one version of a KL-regularized direct win rate optimization objective, we also compare different variants of DWRO-KL. We vary $h$ (identity, log, and logit), $\beta$ (1, 0.1, 0.01, 0.001), and the estimation of the preference classifier (perfect and estimated with and without the Bradley Terry assumption).

### 6.1 EXPERIMENTAL SETUP

We employ Pythia-2.8b (Biderman et al., 2023) as our base model and the OpenAssistant (OASST) (Kopf et al., 2023) and Anthropic Helpfulness and Harmlessness (HH) (Bai et al., 2022) datasets for preference annotations. We train the base models on the data outputs and use these finetuned models as our initial models. To simulate a preference environment, we train an oracle judge model per dataset to estimate $p(\ell = 1 \mid \mathbf{x}, \mathbf{y}_0, \mathbf{y}_1)$ and relabel the preference annotations in the dataset using this judge model. We use this same oracle to evaluate win rate after training. See Appendix I for further details on the judge model. We additionally train 1. a reward model on the oracle-labeled judge annotations (accuracy is 82.8 for OASST and 81.36 for HH) and 2. an imperfect judge model (accuracy is 80.47 for OASST and 85.16 for HH). As expected, the BT assumption is helpful for preference classifier estimation in OASST but not in HH, as the OASST directly abides by this assumption (outputs are globally ranked) whereas HH does not explicitly. For optimization, we use the PPO algorithm implemented in the TRL library (von Werra et al., 2020). See Appendix J for additional training and evaluation details.

**Implementing DWRO-KL.** Different variants of DWRO-KL can be implemented as different choices for the function used to score outputs. Namely, each can be written in the form $\max_\theta \mathbb{E}_{p(\mathbf{x})} \mathbb{E}_{p_\theta(\mathbf{y} \mid \mathbf{x})}[\psi(\mathbf{x}, \mathbf{y}) - \beta \log p_\theta(\mathbf{y} \mid \mathbf{x}) + \beta \log p_{\text{ref}}(\mathbf{y} \mid \mathbf{x})]$ for different choices of $\psi$. All expectations over generator distributions use a single-sample Monte-Carlo estimate. Concretely, for given query we sample one response each from the current and original model to compute the score, which is either a function of 1. the preference probability under the judge model which takes in the pair as input, or 2. the BT preference probability under the reward model which takes each sequence in as input separately. The one exception is DWRO-KL-logit-BT (i.e., RLHF), where we drop $\mathbb{E}_{p(\mathbf{y}_0 \mid \mathbf{x})}[r(\mathbf{x}, \mathbf{y}_0)]$ and only optimize $r(\mathbf{x}, \mathbf{y})$.

### 6.2 RESULTS: COMPARING RLHF, DPO, AND SFT

Figure 1 compares methods across different values of $\beta$ (not applicable for SFT). While SFT performance aligns with expectations, as does DPO performance with more regularization, RLHF substantially underperforms relative to expected given the aforementioned analysis, as does DPO with less regularization ($\beta$=0.001). The non-monotonic nature of both point to the influence of factors beyond target distribution for win rate improvements. To test if RLHF underfperformance is due to

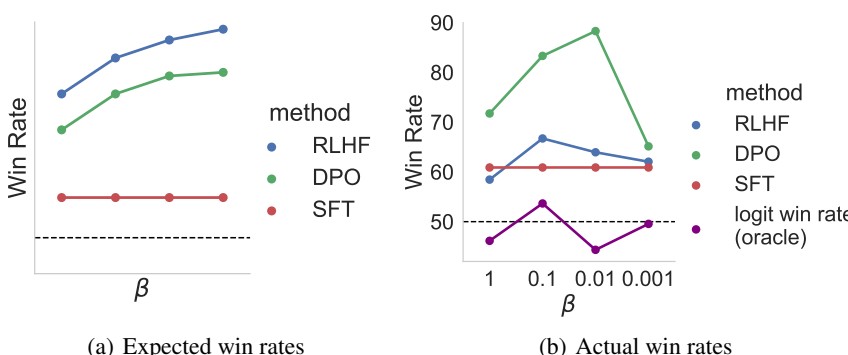

(a) Expected win rates  (b) Actual win rates

Figure 1: Expected versus observed win rates of RLHF, DPO, and SFT over the original model for OASST. RLHF notably underperforms relative to expectations. Moreover, substituting the learned reward model for the oracle preference classifier does not improve performance, suggesting other factors are more important.

Table 1: Win rate results of different DWRO-KL variants over the reference model. Row corresponding to RLHF is shaded grey. No choice of $\hat{p}_\ell$, $h$, or $\beta$ systematically outperforms all others.

| Dataset | $\hat{p}_\ell$ | $h$ | $\beta = 0.001$ | $\beta = 0.01$ | $\beta = 0.1$ | $\beta = 1$ |
|---------|------|------|-----------------|----------------|---------------|-------------|
| HH | non-BT | log | 38.36 (1.71) | 46.38 (1.34) | 63.94 (0.94) | 67.00 (0.83) |
| | | logit | 69.15 (0.91) | 41.98 (1.45) | 48.74 (0.97) | 69.15 (0.93) |
| | | identity | 69.94 (0.99) | 69.94 (0.99) | 65.92 (0.95) | 43.21 (0.54) |
| | BT | log | 35.41 (1.98) | 33.80 (1.59) | 41.76 (1.49) | 49.49 (0.60) |
| | | logit | 70.46 (0.87) | 63.93 (1.24) | 69.44 (0.90) | 54.60 (0.67) |
| | | identity | 65.71 (0.95) | 64.59 (0.95) | 69.94 (0.99) | 51.72 (0.59) |
| | oracle | log | 34.98 (1.74) | 52.74 (0.26) | 47.66 (1.09) | 43.29 (0.65) |
| | | logit | 41.99 (1.65) | 55.13 (1.07) | 50.28 (1.21) | 48.55 (0.44) |
| | | identity | 65.34 (0.93) | 66.42 (1.01) | 68.39 (0.94) | 45.14 (0.44) |
| OASST | non-BT | log | 59.51 (3.88) | 59.02 (3.96) | 61.98 (1.40) | 58.63 (1.98) |
| | | logit | 61.98 (3.53) | 54.15 (3.20) | 65.14 (1.20) | 48.69 (1.33) |
| | | identity | 56.65 (3.26) | 54.20 (2.28) | 64.90 (1.28) | 53.64 (2.10) |
| | BT | log | 63.10 (3.48) | 62.80 (3.55) | 60.94 (3.46) | 54.64 (1.26) |
| | | logit | 62.05 (3.65) | 63.95 (3.38) | 66.72 (1.16) | 58.45 (1.82) |
| | | identity | 62.00 (3.55) | 51.98 (2.96) | 54.29 (2.16) | 50.42 (1.38) |
| | oracle | log | 59.32 (3.79) | 60.19 (3.69) | 56.09 (1.65) | 60.09 (1.21) |
| | | logit | 49.55 (3.06) | 44.34 (2.61) | 53.66 (2.87) | 46.15 (1.55) |
| | | identity | 52.13 (2.47) | 66.69 (1.29) | 66.21 (1.22) | 52.31 (1.25) |

error from using an estimated reward model instead of the true preference classifier, we additionally run a DWRO-logit variant with oracle preference classifier. However, this performs even worse, suggesting that an alternative factor is at play.

## 6.3 RESULTS: COMPARING DWRO-KL VARIANTS

Table 1 compares different DWRO-KL objectives with different choices of $h$, $\beta$, and the estimation of the classifier. Notably, no DWRO-KL method (i.e., choice of $h$) outperforms the others systematically across settings. Moreover, the settings that correspond to better target distributions (i.e., using a perfect preference classifier, low $\beta$, and $h =$logit) do not necessarily yield better win rates empirically. These results suggest that there is a more important consideration than the target distribution implied by the objective, namely the success of optimization. Indeed, we see that training loss correlates more with test win rate across DWRO-KL more so than any of the target distribution design choices $\hat{p}_\ell, h$, or $\beta$ (**p-value of Spearman rank correlation test for train loss vs. win rate is 8.27e-5,**

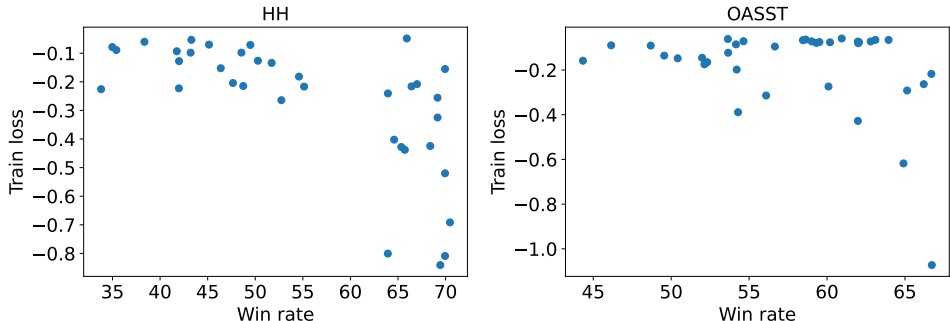

Figure 2: Train loss vs. test win rate across all settings tested in Table 1. Whereas Table 1 does not provide clear evidence that objectives with more favorable target distributions yield better win rates, the above figure shows that the success of optimization is indicative of better win rates.

compared to 0.968, 0.885, and 0.133 for $\hat{p}_\ell, h,$ and $\beta$ respectively). Figure 2 plots train loss vs. test win rate for all DWRO-KL variants. Interestingly, even though the losses for different DWRO variants differ due to choices in $h$ and $\beta$, there still exists a surprisingly noticeable global trend of better loss corresponding to better win rate.

## 7 DISCUSSION

Amidst the increasingly complex landscape around aligning language models to human preferences, this work provides a simple insight: win rate is all that can matter from preference data alone. We first illustrate that the only evaluation grounded in preference data alone is win rate under the preference classifier dictated by the data, which means that the only goal for preference learning that is based in the preference data distribution itself is win rate. This insight prompts us to explain how common procedures in preference learning relate to win rate optimization. We see that, based on their relationship to DWRO, RLHF should be preferred over DPO and SFT. In experiments, however, we see that ease of optimization plays an important role in a method's success. In this regard, SFT is easiest to optimize, then DPO and finally RLHF.

**What's the value of so many preference learning algorithms?** Our analysis may seem to suggest that there is little benefit in developing many different preference learning methods; after all, directly optimizing for win rate seems to be the theoretical ideal, and different choices for DWRO objectives do not seem to currently play a significant role empirically. However, the combined overall theoretical and empirical analysis paint a different picture: notably, no existing method is optimal with respect to both the objective being optimized and the ease of optimization, suggesting why it might be useful to combine methods (i.e., to take advantage of different strengths) as well as develop new ones (i.e., to strike a better overall balance).

**What's next for preference learning?** This work offers several takeaways for future work. First and foremost, the analysis suggests that the most important improvements in preference learning will likely fall under the umbrella of moving closer directly optimizing for win rate, either in the objective itself or the practical optimization of it. How might we be able to make progress on either front? For one, the connection between RLHF / DWRO-KL objectives and variational inference suggests a rich field of inquiry to draw upon from the latter (variational inference, probabilistic inference) to improve the former (preference learning), from variance reduction techniques (Mohamed et al., 2020b) to alternative optimization objectives and algorithms altogether (e.g. (Naesseth et al., 2020)). There are likely many other promising directions for future inquiry; as long as future work focuses on connecting to the central goal of (better) win rate optimization, there is amply opportunity to advance the endeavor of learning from preference data.

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

## A    PROPOSITION 1 PROOF

**Proposition 1.** *Under Definition 2, $\phi$ must be*

$$\phi_{p(\mathbf{y}_0 \mid \boldsymbol{x})}(p(\mathbf{y} \mid \boldsymbol{x}), \mathcal{E}) = \mathbb{E}_{p(\boldsymbol{x})}\mathbb{E}_{p(\boldsymbol{y} \mid \boldsymbol{x})}\mathbb{E}_{p(\mathbf{y}_0 \mid \boldsymbol{x})}[h \cdot p(\ell = 1 \mid \boldsymbol{x}, \mathbf{y}_0, \boldsymbol{y})] \tag{15}$$

*for some choice of $p(\mathbf{y}_0 \mid \boldsymbol{x})$.*

*Proof.* Property 1 forces $\phi$ to be a functional of the preference classifier, while Property 2 forces the preference classifications to be aggregated linearly across query environment, generator, and anchor. Starting from the base case implied by Property 1, we can define $\phi$ for query, generator, and anchor distributions which are each a linear combination of two Dirac deltas. Let $p(\mathbf{x}) = ap_1(\mathbf{x}) + bp_2(\mathbf{x})$, $p(\mathbf{y} \mid \mathbf{x}) = c_{\mathbf{x}}p_1(\mathbf{y} \mid \mathbf{x}) + d_{\mathbf{x}}p_2(\mathbf{y} \mid \mathbf{x})$ for each $\mathbf{x}$, and $p(\mathbf{y}_0 \mid \mathbf{x}) = e_{\mathbf{x}}p_1(\mathbf{y}_0 \mid \mathbf{x}) + f_{\mathbf{x}}p_2(\mathbf{y}_0 \mid \mathbf{x})$; assume all constants yield normalized probability distributions. Then,

$$\phi_{p(\mathbf{y}_0 \mid \mathbf{x})}(p(\mathbf{y} \mid \mathbf{x}), \mathcal{E}) = a\phi_{p(\mathbf{y}_0 \mid \mathbf{x})}(p(\mathbf{y} \mid \mathbf{x}), \mathcal{E}_1) + b\phi_{p(\mathbf{y}_0 \mid \mathbf{x})}(p(\mathbf{y} \mid \mathbf{x}), \mathcal{E}_2) \tag{16}$$

$$= a[c_{\mathbf{x}_a}\phi_{p(\mathbf{y}_0 \mid \mathbf{x})}(p_1(\mathbf{y} \mid \mathbf{x}), \mathcal{E}_1) + d_{\mathbf{x}_a}\phi_{p(\mathbf{y}_0 \mid \mathbf{x})}(p_2(\mathbf{y} \mid \mathbf{x}), \mathcal{E}_1)]$$

$$+ b[c_{\mathbf{x}_b}\phi_{p(\mathbf{y}_0 \mid \mathbf{x})}(p_1(\mathbf{y} \mid \mathbf{x}), \mathcal{E}_2) + d_{\mathbf{x}_b}\phi_{p(\mathbf{y}_0 \mid \mathbf{x})}(p_1(\mathbf{y} \mid \mathbf{x}), \mathcal{E}_2)] \tag{17}$$

$$= a[c_{\mathbf{x}_a}[e_{\mathbf{x}_a}\phi_{p_1(\mathbf{y}_0 \mid \mathbf{x})}(p_1(\mathbf{y} \mid \mathbf{x}), \mathcal{E}_1) + f_{\mathbf{x}_a}\phi_{p_2(\mathbf{y}_0 \mid \mathbf{x})}(p_1(\mathbf{y} \mid \mathbf{x}), \mathcal{E}_1)]$$

$$+ d_{\mathbf{x}_a}[e_{\mathbf{x}_a}\phi_{p(\mathbf{y}_0 \mid \mathbf{x})}(p_2(\mathbf{y} \mid \mathbf{x}), \mathcal{E}_1) + f_{\mathbf{x}_a}\phi_{p(\mathbf{y}_0 \mid \mathbf{x})}(p_2(\mathbf{y} \mid \mathbf{x}), \mathcal{E}_2)]]$$

$$+ b[c_{\mathbf{x}_b}[e_{\mathbf{x}_b}\phi_{p_1(\mathbf{y}_0 \mid \mathbf{x})}(p_1(\mathbf{y} \mid \mathbf{x}), \mathcal{E}_1) + f_{\mathbf{x}_b}\phi_{p_2(\mathbf{y}_0 \mid \mathbf{x})}(p_1(\mathbf{y} \mid \mathbf{x}), \mathcal{E}_1)]$$

$$+ d_{\mathbf{x}_b}[e_{\mathbf{x}_b}\phi_{p(\mathbf{y}_0 \mid \mathbf{x})}(p_2(\mathbf{y} \mid \mathbf{x}), \mathcal{E}_1) + f_{\mathbf{x}_b}\phi_{p_2(\mathbf{y}_0 \mid \mathbf{x})}(p_2(\mathbf{y} \mid \mathbf{x}), \mathcal{E}_2)]]$$

$$= \sum_{i \in \{a,b\}} \sum_{j \in \{c_i,d_i\}} \sum_{k \in \{e_i,f_i\}} ijk(h \cdot \phi(\ell = 1 \mid \mathbf{x}_i, \mathbf{y}_j, \mathbf{y}_k)). \tag{18}$$

Generalizing to any discrete distribution for query, generator, and anchor, we have Equation (4). □

## B    PROPOSITION 2 PROOF

We provide both the informal statement in the main paper as well as its formal version.

**Proposition 2.** (informal) *Under the Bradley-Terry assumption, all DWRO objectives with monotonic $h$ share the same optimal solution.*

**Proposition 2.** (formal) *Denote by $\mathcal{P}_h^*$ the set of distributions $p(\mathbf{y} \mid \boldsymbol{x})$ that optimize the DWRO-$h$ objective. Assume a hypothesis class induced by $\theta \in \Theta$ such that all optima are realizable. Then, for a given anchor distribution $p(\mathbf{y}_0 \mid \boldsymbol{x})$, $\mathcal{P}_h^* = \mathcal{P}_{h'}^*$ for any monotonic $h$ under the Bradley-Terry assumption with finite rewards.*

*Proof.* We first introduce the $\mathcal{P}_{\text{reward}}^*$, the set of all distributions $p(\mathbf{y} \mid \mathbf{x})$ which for each $\mathbf{x}$ place all their probability mass over only the highest-reward sequences or some subset of them. Then, we show that $\mathcal{P}_{\text{reward}}^* = \mathcal{P}_h^*$ for any monotonic $h$ and any anchor distribution $p(\mathbf{y}_0 \mid \mathbf{x})$. In other words, any maximum-reward distribution $p_{\text{reward}}^*(\mathbf{y} \mid \mathbf{x}) \in \mathcal{P}_{\text{reward}}^*$ maximizes any DWRO objective and vice versa:

$$\mathbb{E}_{p(\mathbf{x})}\mathbb{E}_{p_{\text{reward}}^*(\mathbf{y} \mid \mathbf{x})}\mathbb{E}_{p(\mathbf{y}_0 \mid \mathbf{x})}[h \circ p(\ell = 1 \mid \mathbf{x}, \mathbf{y}_0, \mathbf{y})] \tag{19}$$

$$= \mathbb{E}_{p(\mathbf{x})}\mathbb{E}_{p_{\text{reward}}^*(\mathbf{y} \mid \mathbf{x})}\mathbb{E}_{p(\mathbf{y}_0 \mid \mathbf{x})}[h \circ \sigma\left(r(\mathbf{x}, \mathbf{y}) - r(\mathbf{x}, \mathbf{y}_0)\right)] \tag{20}$$

$$= \max_\theta \mathbb{E}_{p(\mathbf{x})}\mathbb{E}_{p_\theta(\mathbf{y}_1 \mid \mathbf{x})}\mathbb{E}_{p(\mathbf{y}_0 \mid \mathbf{x})}[h \circ \sigma\left(r(\mathbf{x}, \mathbf{y}_1) - r(\mathbf{x}, \mathbf{y}_0)\right)] \tag{21}$$

$$= \max_\theta \mathbb{E}_{p(\mathbf{x})}\mathbb{E}_{p_\theta(\mathbf{y}_1 \mid \mathbf{x})}\mathbb{E}_{p(\mathbf{y}_0 \mid \mathbf{x})}[h \circ p(\ell = 1 \mid \mathbf{x}, \mathbf{y}_0, \mathbf{y}_1)]. \tag{22}$$

It follows that $\mathcal{P}_h^* = \mathcal{P}_{h'}^*$ for any monotonic $h$. □

## C  Target distribution derivations

### C.0.1  DWRO-KL

Here, we show that the DWRO-KL objective is equivalent to minimizing the reverse KL divergence of the model and following target distribution:

$$p^*_{\text{DWRO-KL}}(\mathbf{y}\,|\,\mathbf{x}) \propto p_{\text{ref}}(\mathbf{y}\,|\,\mathbf{x}) \exp(\frac{1}{\beta}\mathbb{E}_{p(\mathbf{y}_0|\mathbf{x})}[h \cdot p(\ell = 1|\mathbf{x}, \mathbf{y}_0, \mathbf{y})]). \tag{23}$$

Derivation:

$$\max_{\theta} \mathbb{E}_{p(\mathbf{x})}\left[\mathbb{E}_{p_\theta(\mathbf{y}_1\,|\,\mathbf{x})}\mathbb{E}_{p(\mathbf{y}_0\,|\,\mathbf{x})}[h \cdot p(\ell = 1\,|\,\mathbf{x}, \mathbf{y}_0, \mathbf{y}_1)] - \beta\text{KL}(p_\theta(\mathbf{y}\,|\,\mathbf{x}) \,\|\, p_{\text{ref}}(\mathbf{y}\,|\,\mathbf{x}))\right] \tag{24}$$

$$= \min_{\theta} -\mathbb{E}_{p(\mathbf{x})}\mathbb{E}_{p_\theta(\mathbf{y}_1\,|\,\mathbf{x})}\left[\mathbb{E}_{p(\mathbf{y}_0\,|\,\mathbf{x})}[h \cdot p(\ell = 1\,|\,\mathbf{x}, \mathbf{y}_0, \mathbf{y}_1)] - \beta \log \frac{p_\theta(\mathbf{y}\,|\,\mathbf{x})}{p_{\text{ref}}(\mathbf{y}\,|\,\mathbf{x})}\right] \tag{25}$$

$$= \min_{\theta} \mathbb{E}_{p(\mathbf{x})}\mathbb{E}_{p_\theta(\mathbf{y}_1\,|\,\mathbf{x})}\left[\log \frac{p_\theta(\mathbf{y}\,|\,\mathbf{x})}{p_{\text{ref}}(\mathbf{y}\,|\,\mathbf{x})} - \frac{1}{\beta}\mathbb{E}_{p(\mathbf{y}_0\,|\,\mathbf{x})}[h \cdot p(\ell = 1\,|\,\mathbf{x}, \mathbf{y}_0, \mathbf{y}_1)]\right] \tag{26}$$

$$= \min_{\theta} \mathbb{E}_{p(\mathbf{x})}\mathbb{E}_{p_\theta(\mathbf{y}_1\,|\,\mathbf{x})}\left[\log \frac{p_\theta(\mathbf{y}\,|\,\mathbf{x})}{p_{\text{ref}}(\mathbf{y}\,|\,\mathbf{x}) \exp(\frac{1}{\beta}\mathbb{E}_{p(\mathbf{y}_0\,|\,\mathbf{x})}[h \cdot p(\ell = 1\,|\,\mathbf{x}, \mathbf{y}_0, \mathbf{y}_1)])}\right] \tag{27}$$

$$= \min_{\theta} \mathbb{E}_{p(\mathbf{x})}\mathbb{E}_{p_\theta(\mathbf{y}_1\,|\,\mathbf{x})}\left[\log \frac{p_\theta(\mathbf{y}\,|\,\mathbf{x})}{\frac{1}{Z(\mathbf{x})}p_{\text{ref}}(\mathbf{y}\,|\,\mathbf{x}) \exp(\frac{1}{\beta}\mathbb{E}_{p(\mathbf{y}_0\,|\,\mathbf{x})}[h \cdot p(\ell = 1\,|\,\mathbf{x}, \mathbf{y}_0, \mathbf{y}_1)])} - \log Z(\mathbf{x})\right] \tag{28}$$

$$= \min_{\theta} \mathbb{E}_{p(\mathbf{x})}\left[\text{KL}(p_\theta(\mathbf{y}\,|\,\mathbf{x}) \,\|\, p^*_{\text{DWRO-KL}}(\mathbf{y}\,|\,\mathbf{x}))\right], \tag{29}$$

$$p^*_{\text{DWRO-KL}}(\mathbf{y}\,|\,\mathbf{x}) \propto p_{\text{ref}}(\mathbf{y}\,|\,\mathbf{x}) \exp\left(\frac{1}{\beta}\mathbb{E}_{p(\mathbf{y}_0|\mathbf{x})}[h \cdot p(\ell = 1|\mathbf{x}, \mathbf{y}_0, \mathbf{y})]\right). \tag{30}$$

### C.0.2  RLHF

Here, we show that, under the BT assumption, the RLHF objective is equivalent to minimizing the reverse KL divergence with the following target distribution:

$$p^*_{\text{RLHF}}(\mathbf{y}\,|\,\mathbf{x}) \propto p_{\text{ref}}(\mathbf{y}\,|\,\mathbf{x}) \exp(\frac{1}{\beta}\mathbb{E}_{p(\mathbf{y}_0|\mathbf{x})}[\text{logit } p(\ell = 1|\mathbf{x}, \mathbf{y}_0, \mathbf{y})]). \tag{31}$$

$$\max_{\theta} \mathbb{E}_{p(\mathbf{x})}\left[\mathbb{E}_{p_\theta(\mathbf{y}\,|\,\mathbf{x})}[r(\mathbf{x}, \mathbf{y})] - \beta\text{KL}(p_\theta(\mathbf{y}\,|\,\mathbf{x}) \,\|\, p_{\text{ref}}(\mathbf{y}\,|\,\mathbf{x}))\right] \tag{32}$$

$$= \max_{\theta} \mathbb{E}_{p(\mathbf{x})}\left[\mathbb{E}_{p_\theta(\mathbf{y}\,|\,\mathbf{x})}\mathbb{E}_{p(\mathbf{y}_0\,|\,\mathbf{x})}[\text{logit } p(\ell = 1\,|\,\mathbf{x}, \mathbf{y}_0, \mathbf{y})] - \beta\text{KL}(p_\theta(\mathbf{y}\,|\,\mathbf{x}) \,\|\, p_{\text{ref}}(\mathbf{y}\,|\,\mathbf{x}))\right] \tag{33}$$

$$= \min_{\theta} \mathbb{E}_{p(\mathbf{x})}\mathbb{E}_{p_\theta(\mathbf{y}\,|\,\mathbf{x})}\left[\log \frac{(p_\theta(\mathbf{y}\,|\,\mathbf{x})}{p_{\text{ref}}(\mathbf{y}\,|\,\mathbf{x}))} - \frac{1}{\beta}\mathbb{E}_{p(\mathbf{y}_0\,|\,\mathbf{x})}[\text{logit } p(\ell = 1\,|\,\mathbf{x}, \mathbf{y}_0, \mathbf{y})]\right] \tag{34}$$

$$= \min_{\theta} \mathbb{E}_{p(\mathbf{x})}\left[\text{KL}(p_\theta(\mathbf{y}\,|\,\mathbf{x}) \,\|\, p^*_{\text{RLHF}}(\mathbf{y}\,|\,\mathbf{x}))\right], \tag{35}$$

$$p^*_{\text{RLHF}}(\mathbf{y}\,|\,\mathbf{x}) \propto p(\mathbf{y}\,|\,\mathbf{x}) \exp\left(\frac{1}{\beta}\mathbb{E}_{p(\mathbf{y}_0|\mathbf{x})}[\text{logit } p(\ell = 1|\mathbf{x}, \mathbf{y}_0, \mathbf{y})]\right). \tag{36}$$

### C.0.3  Reverse-KL objective for SFT Target distribution

Here, we show that the objective in Equation (11) is equivalent to optimizing for the SFT target (Equation (12)) via minimizing the reverse KL divergence.

$$\max_{\theta} \mathbb{E}_{p(\mathbf{x})}\left[\mathbb{E}_{p_\theta(\mathbf{y}_1\,|\,\mathbf{x})} \log \mathbb{E}_{p(\mathbf{y}_0\,|\,\mathbf{x})}[p(\ell = 1\,|\,\mathbf{x}, \mathbf{y}_0, \mathbf{y}_1)] - \text{KL}(p_\theta(\mathbf{y}\,|\,\mathbf{x}) \,\|\, p_{\text{ref}}(\mathbf{y}\,|\,\mathbf{x}))\right] \tag{37}$$

$$= \min_{\theta} \mathbb{E}_{p(\mathbf{x})}\mathbb{E}_{p_\theta(\mathbf{y}_1\,|\,\mathbf{x})}\left[\log \frac{p_\theta(\mathbf{y}\,|\,\mathbf{x})}{p_{\text{ref}}(\mathbf{y}\,|\,\mathbf{x})} - \log \mathbb{E}_{p(\mathbf{y}_0\,|\,\mathbf{x})}[p(\ell = 1\,|\,\mathbf{x}, \mathbf{y}_0, \mathbf{y}_1)]\right] \tag{38}$$

$$= \min_{\theta} \mathbb{E}_{p(\mathbf{x})}\left[\text{KL}(p_\theta(\mathbf{y}\,|\,\mathbf{x}) \,\|\, p^*_{\text{SFT}}(\mathbf{y}\,|\,\mathbf{x}))\right], \tag{39}$$

$$p^*_{\text{SFT}}(\mathbf{y}\,|\,\mathbf{x}) \propto p(\mathbf{y}\,|\,\mathbf{x})\mathbb{E}_{p(\mathbf{y}_0|\mathbf{x})}[p(\ell = 1|\mathbf{x}, \mathbf{y}_0, \mathbf{y})]. \tag{40}$$

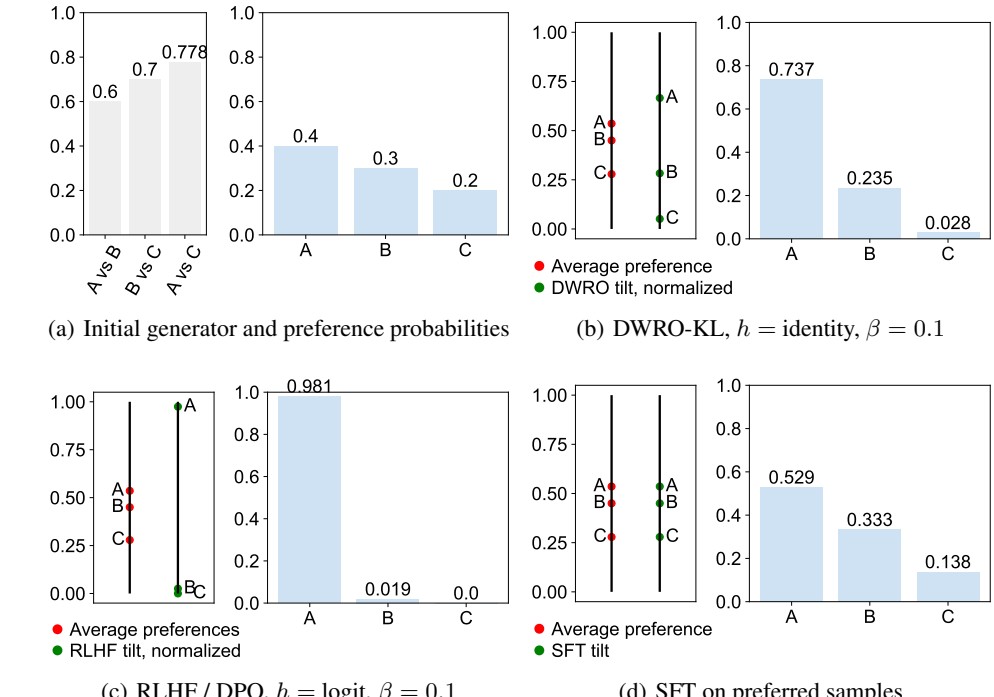

(a) Initial generator and preference probabilities

(b) DWRO-KL, $h = $ identity, $\beta = 0.1$

(c) RLHF / DPO, $h = $ logit, $\beta = 0.1$

(d) SFT on preferred samples

Figure 3: Different preference learning objectives have different target distributions. Consider the initial setting in (a). DWRO-KL with $h = $ identity, $\beta = 0.1$ yields the target distribution depicted in (b). RLHF ($h = $ logit, $\beta = 0.1$) yields a sharper distribution given that average logit probabilities can differ more from each other than average probabilities (c). SFT yields the least sharp distribution as it can only apply weights between 0 and 1 (d).

## D  COMPARING TARGET DISTRIBUTIONS

In Figure 3, we present a visualization of the target distributions of different preference learning objectives. Panel (a) shows an example preference environment defined by the true preference probabilities between responses (left) and initial starting model (right). Panels (b), (c), and (d) visualize the resulting target distribution of different objectives. In each, the figure on the left shows how the average preference probabilities $\mathbb{E}_{p(\mathbf{y}_0 \mid \mathbf{x})}[p(\ell = 1 \mid \mathbf{x}, \mathbf{y}_0, \mathbf{y}_1]$ (red dots) are translated into the tilt applied to the starting model, i.e. $g(\mathbf{y}, \mathbf{x})$ in $p^*(\mathbf{y} \mid \mathbf{x}) \propto p(\mathbf{y} \mid \mathbf{x}) g(\mathbf{y}, \mathbf{x})$ (green dots). The figure on the right shows the optimal distribution under the objective. Notably, among the DWRO-KL family, the choice of $h$ can make a substantial difference in the target distribution of the objective (panels b vs. c). Moreover, SFT is limited in how much mass it can put on the preferred sample A, as $g(\mathbf{y}, \mathbf{x})$ is only the average preference probabilities themselves (panel d).

## E  THEOREM 1 PROOF

We first provide results for a more general setting (Lemma 1); then we specialize to the setting in the main paper.

**Lemma 1.** *Let $p(\mathbf{y}_0 \mid \mathbf{x})$ be the initial generative model, and $p_{SFT}(\mathbf{y} \mid \mathbf{x})$ be the target distribution of supervised finetuning on preferred samples $\mathbf{y}_0, \mathbf{y}_1 \sim p(\mathbf{y}_0, \mathbf{y}_1 \mid \mathbf{x}) = p(\mathbf{y}_0 \mid \mathbf{x})p(\mathbf{y}_1 \mid \mathbf{x})$. Then,*

$$Win\ Rate_{p(\mathbf{y}_0 \mid \mathbf{x})}[p_{SFT}(\mathbf{y} \mid \mathbf{x})] = \tag{41}$$

$$Win\ Rate_{p(\mathbf{y}_0 \mid \mathbf{x})}[p(\mathbf{y}_1 \mid \mathbf{x})] + \int p(\mathbf{x}) \left[ \frac{Variance_{p(\mathbf{y}_1 \mid \mathbf{x})} \left[ \int p(\mathbf{y}_0 \mid \mathbf{x})p(\ell = 1 \mid \mathbf{x}, \mathbf{y}_0, \mathbf{y}_1) d\mathbf{y}_0 \right]}{\int p(\mathbf{y}_1' \mid \mathbf{x}) \int p(\ell = 1 | \mathbf{x}, \mathbf{y}_0'', \mathbf{y}_1') p(\mathbf{y}_0 \mid \mathbf{x}) d\mathbf{y}_0'' d\mathbf{y}_1'} \right] d\mathbf{x}.$$

$$\tag{42}$$

*Proof.*

$$\text{Win Rate}_{p(\mathbf{y}\,|\,\mathbf{x})}[p_{\text{SFT}}(\mathbf{y}\,|\,\mathbf{x})] = \int p(\mathbf{x})p(\mathbf{y}_0\,|\,\mathbf{x})p_{\text{SFT}}(\mathbf{y}\,|\,\mathbf{x})p(\ell=1\,|\,\mathbf{x},\mathbf{y}_0,\mathbf{y}_1)\mathbf{y}_0\mathbf{y}_1 d\mathbf{x} \tag{43}$$

$$= \int p(\mathbf{x})p(\mathbf{y}_0\,|\,\mathbf{x})\frac{p(\mathbf{y}_1\,|\,\mathbf{x})\int p(\ell=1|\mathbf{x},\mathbf{y}_0',\mathbf{y}_1)p(\mathbf{y}_0'\,|\,\mathbf{x})d\mathbf{y}_0'}{\int p(\mathbf{y}_1'\,|\,\mathbf{x})\int p(\ell=1|\mathbf{x},\mathbf{y}_0'',\mathbf{y}_1')p(\mathbf{y}_0''\,|\,\mathbf{x})d\mathbf{y}_0''d\mathbf{y}_1'}p(\ell=1\,|\,\mathbf{x},\mathbf{y}_0,\mathbf{y}_1)d\mathbf{y}_0 d\mathbf{y}_1 d\mathbf{x} \tag{44}$$

$$= \int p(\mathbf{x})\left[p(\mathbf{y}_0\,|\,\mathbf{x})\frac{p(\mathbf{y}_1\,|\,\mathbf{x})\int p(\ell=1|\mathbf{x},\mathbf{y}_0',\mathbf{y}_1)p(\mathbf{y}_0'\,|\,\mathbf{x})d\mathbf{y}_0'}{\int p(\mathbf{y}_1'\,|\,\mathbf{x})\int p(\ell=1|\mathbf{x},\mathbf{y}_0'',\mathbf{y}_1')p(\mathbf{y}_0''\,|\,\mathbf{x})d\mathbf{y}_0''d\mathbf{y}_1'}p(\ell=1\,|\,\mathbf{x},\mathbf{y}_0,\mathbf{y}_1)d\mathbf{y}_0 d\mathbf{y}_1\right]d\mathbf{x} \tag{45}$$

$$= \int p(\mathbf{x})\left[\int p(\mathbf{y}_0\,|\,\mathbf{x})\frac{p(\mathbf{y}_1\,|\,\mathbf{x})\int p(\ell=1|\mathbf{x},\mathbf{y}_0',\mathbf{y}_1)p(\mathbf{y}_0'\,|\,\mathbf{x})d\mathbf{y}_0'}{\int p(\mathbf{y}_1'\,|\,\mathbf{x})\int p(\ell=1|\mathbf{x},\mathbf{y}_0'',\mathbf{y}_1')p(\mathbf{y}_0''\,|\,\mathbf{x})d\mathbf{y}_0''d\mathbf{y}_1'}p(\ell=1\,|\,\mathbf{x},\mathbf{y}_0,\mathbf{y}_1)d\mathbf{y}_0 d\mathbf{y}_1\right]d\mathbf{x} \tag{46}$$

$$= \int p(\mathbf{x})\left[\int \frac{\int p(\mathbf{y}_0\,|\,\mathbf{x})p(\ell=1\,|\,\mathbf{x},\mathbf{y}_0,\mathbf{y}_1)d\mathbf{y}_0 p(\mathbf{y}_1\,|\,\mathbf{x})\int p(\ell=1|\mathbf{x},\mathbf{y}_0',\mathbf{y}_1)p(\mathbf{y}_0'\,|\,\mathbf{x})d\mathbf{y}_0'}{\int p(\mathbf{y}_1'\,|\,\mathbf{x})\int p(\ell=1|\mathbf{x},\mathbf{y}_0'',\mathbf{y}_1')p(\mathbf{y}_0\,|\,\mathbf{x})d\mathbf{y}_0''d\mathbf{y}_1'}d\mathbf{y}_1\right]d\mathbf{x} \tag{47}$$

$$= \int p(\mathbf{x})\left[\frac{\int(\int p(\mathbf{y}_0\,|\,\mathbf{x})p(\ell=1\,|\,\mathbf{x},\mathbf{y}_0,\mathbf{y}_1)d\mathbf{y}_0)^2 p(\mathbf{y}_1\,|\,\mathbf{x})d\mathbf{y}_1}{\int p(\mathbf{y}_1'\,|\,\mathbf{x})\int p(\ell=1|\mathbf{x},\mathbf{y}_0'',\mathbf{y}_1')p(\mathbf{y}_0\,|\,\mathbf{x})d\mathbf{y}_0''d\mathbf{y}_1'}\right]d\mathbf{x} \tag{48}$$

$$= \int p(\mathbf{x})\left[\frac{\left(\int(\int p(\mathbf{y}_0\,|\,\mathbf{x})p(\ell=1\,|\,\mathbf{x},\mathbf{y}_0,\mathbf{y}_1)d\mathbf{y}_0)p(\mathbf{y}_1\,|\,\mathbf{x})d\mathbf{y}_1\right)^2}{\int p(\mathbf{y}_1'\,|\,\mathbf{x})\int p(\ell=1|\mathbf{x},\mathbf{y}_0'',\mathbf{y}_1')p(\mathbf{y}_0\,|\,\mathbf{x})d\mathbf{y}_0''d\mathbf{y}_1'}\right]d\mathbf{x} \tag{49}$$

$$+ \int p(\mathbf{x})\left[\frac{\text{Variance}_{p(\mathbf{y}_1\,|\,\mathbf{x})}\left[\int p(\mathbf{y}_0\,|\,\mathbf{x})p(\ell=1\,|\,\mathbf{x},\mathbf{y}_0,\mathbf{y}_1)d\mathbf{y}_0)\right]}{\int p(\mathbf{y}_1'\,|\,\mathbf{x})\int p(\ell=1|\mathbf{x},\mathbf{y}_0'',\mathbf{y}_1')p(\mathbf{y}_0\,|\,\mathbf{x})d\mathbf{y}_0''d\mathbf{y}_1'}\right]d\mathbf{x} \tag{50}$$

$$= \int p(\mathbf{x})(\int p(\mathbf{y}_0\,|\,\mathbf{x})p(\ell=1\,|\,\mathbf{x},\mathbf{y}_0,\mathbf{y}_1)d\mathbf{y}_0)p(\mathbf{y}_1\,|\,\mathbf{x})d\mathbf{y}_1 d\mathbf{x} \tag{51}$$

$$+ \int p(\mathbf{x})\left[\frac{\text{Variance}_{p(\mathbf{y}_1\,|\,\mathbf{x})}\left[\int p(\mathbf{y}_0\,|\,\mathbf{x})p(\ell=1\,|\,\mathbf{x},\mathbf{y}_0,\mathbf{y}_1)d\mathbf{y}_0)\right]}{\int p(\mathbf{y}_1'\,|\,\mathbf{x})\int p(\ell=1|\mathbf{x},\mathbf{y}_0'',\mathbf{y}_1')p(\mathbf{y}_0\,|\,\mathbf{x})d\mathbf{y}_0''d\mathbf{y}_1'}\right]d\mathbf{x} \tag{52}$$

$$= \text{Win Rate}_{p(\mathbf{y}_0\,|\,\mathbf{x})}[p(\mathbf{y}_1\,|\,\mathbf{x})] + \int p(\mathbf{x})\left[\frac{\text{Variance}_{p(\mathbf{y}_1\,|\,\mathbf{x})}\left[\int p(\mathbf{y}_0\,|\,\mathbf{x})p(\ell=1\,|\,\mathbf{x},\mathbf{y}_0,\mathbf{y}_1)d\mathbf{y}_0)\right]}{\int p(\mathbf{y}_1'\,|\,\mathbf{x})\int p(\ell=1|\mathbf{x},\mathbf{y}_0'',\mathbf{y}_1')p(\mathbf{y}_0\,|\,\mathbf{x})d\mathbf{y}_0''d\mathbf{y}_1'}\right]d\mathbf{x}. \tag{53}$$

$\square$

**Theorem 1.** (Win rate improvement of SFT) *Let $p(\mathbf{y}_0\,|\,\boldsymbol{x})$ be the initial generative model, and $p_{SFT}(\boldsymbol{y}\,|\,\boldsymbol{x})$ be the target distribution of supervised finetuning on preferred samples ($p(\mathbf{y}_1\,|\,\boldsymbol{x},\ell=1)$, $p(\mathbf{y}_0\,|\,\boldsymbol{x})=p(\mathbf{y}_1\,|\,\boldsymbol{x})$). Then,*

$$\text{Win Rate}_{p(\mathbf{y}_0\,|\,\boldsymbol{x})}[p_{SFT}(\boldsymbol{y}\,|\,\boldsymbol{x})] = 0.5 + 2\mathbb{E}_{p(\mathbf{x})}Var_{p(\mathbf{y}_1\,|\,\boldsymbol{x})}\left[\int p(\mathbf{y}_0\,|\,\boldsymbol{x})p(\ell=1\,|\,\boldsymbol{x},\mathbf{y}_0,\mathbf{y}_1)d\mathbf{y}_0\right], \tag{54}$$

*which is less than 1.0 as long as there exist non-deterministic preference probabilities, $p(\ell=1\,|\,\boldsymbol{x},\mathbf{y}_0,\mathbf{y}_1) \in (0,1)$.*

*Proof.* We use the result of Lemma 1 and plug in the condition $p(\mathbf{y}_0\,|\,\mathbf{x})=p(\mathbf{y}_1\,|\,\mathbf{x})$:

$$\text{Win Rate}_{p(\mathbf{y}_0\,|\,\mathbf{x})}[p_{\text{SFT}}(\mathbf{y}\,|\,\mathbf{x})] \tag{55}$$

$$= 0.5 + \int p(\mathbf{x})\left[\frac{\text{Variance}_{p(\mathbf{y}_1\,|\,\mathbf{x})}\left[\int p(\mathbf{y}_0\,|\,\mathbf{x})p(\ell=1\,|\,\mathbf{x},\mathbf{y}_0,\mathbf{y}_1)d\mathbf{y}_0)\right]}{\int p(\mathbf{y}_1'\,|\,\mathbf{x})\int p(\ell=1|\mathbf{x},\mathbf{y}_0'',\mathbf{y}_1')p(\mathbf{y}_0\,|\,\mathbf{x})d\mathbf{y}_0''d\mathbf{y}_1'}\right]d\mathbf{x} \tag{56}$$

$$= 0.5 + 2\int p(\mathbf{x})\text{Variance}_{p(\mathbf{y}_1\,|\,\mathbf{x})}\left[\int p(\mathbf{y}_0\,|\,\mathbf{x})p(\ell=1\,|\,\mathbf{x},\mathbf{y}_0,\mathbf{y}_1)d\mathbf{y}_0)\right]d\mathbf{x}. \tag{57}$$

Then, $\int p(\mathbf{y}_0 \mid \mathbf{x}) p(\ell = 1 \mid \mathbf{x}, \mathbf{y}_0, \mathbf{y}_1) d\mathbf{y}_0$ can only take values between 0 and 1. The maximal variance of a random variable supported between 0 and 1 is $1/4$, achieved by placing mass equally on only the endpoints 0 and 1. Any non-deterministic preference probability $p(\ell = 1 \mid \mathbf{x}, \mathbf{y}_0, \mathbf{y}_1)$ will yield $\int p(\mathbf{y}_0 \mid \mathbf{x}) p(\ell = 1 \mid \mathbf{x}, \mathbf{y}_0, \mathbf{y}_1) d\mathbf{y}_0 \in (0, 1)$, resulting in a variance less than $1/4$ and a resulting win rate less than 1.0. $\qquad\square$

## F    EXPECTED WIN RATE IMPROVEMENT EXPRESSIONS

Below, we present the expected win rate improvement over the original model for DWRO-KL objectives. Letting $p(\mathbf{y}_0 \mid \mathbf{x}) = p(\mathbf{y}_0' \mid \mathbf{x}) = p(\mathbf{y}_0'' \mid \mathbf{x})$ and $p(\mathbf{y}_1 \mid \mathbf{x}) = p(\mathbf{y}_1' \mid \mathbf{x})$, we have:

$$\text{Win Rate}_{p(\mathbf{y}_0 \mid \mathbf{x})}[p_{\text{DWRO-KL}}(\mathbf{y} \mid \mathbf{x})]$$

$$= \int p(\mathbf{x}) \left[ \frac{\mathbb{E}_{p(\mathbf{y}_1 \mid \mathbf{x})}\left[\mathbb{E}_{p(\mathbf{y}_0 \mid \mathbf{x})} p(\ell = 1 \mid \mathbf{x}, \mathbf{y}_0, \mathbf{y}_1) \exp\left(\frac{1}{\beta}\mathbb{E}_{p(\mathbf{y}_0' \mid \mathbf{x})} h \cdot p(\ell = 1 \mid \mathbf{x}, \mathbf{y}_0', \mathbf{y}_1)\right)\right]}{\mathbb{E}_{p(\mathbf{y}_1' \mid \mathbf{x})} \exp\left(\frac{1}{\beta}\mathbb{E}_{p(\mathbf{y}_0'' \mid \mathbf{x})} h \cdot p(\ell = 1 \mid \mathbf{x}, \mathbf{y}_0'', \mathbf{y}_1')\right)} \right] d\mathbf{x}. \tag{58}$$

Consequently, assuming the BT assumption holds in the preference environment, the expected win rate improvement for the target of RLHF/DPO is as follows:

$$\text{Win Rate}_{p(\mathbf{y}_0 \mid \mathbf{x})}[p_{\text{RLHF/DPO}}(\mathbf{y} \mid \mathbf{x})]$$

$$= \int p(\mathbf{x}) \left[ \frac{\mathbb{E}_{p(\mathbf{y}_1 \mid \mathbf{x})}\left[\mathbb{E}_{p(\mathbf{y}_0 \mid \mathbf{x})} p(\ell = 1 \mid \mathbf{x}, \mathbf{y}_0, \mathbf{y}_1) \exp\left(\frac{1}{\beta}\mathbb{E}_{p(\mathbf{y}_0' \mid \mathbf{x})}[\text{logit } p(\ell = 1 \mid \mathbf{x}, \mathbf{y}_0', \mathbf{y}_1)]\right)\right]}{\mathbb{E}_{p(\mathbf{y}_1' \mid \mathbf{x})} \exp\left(\frac{1}{\beta}\mathbb{E}_{p(\mathbf{y}_0'' \mid \mathbf{x})}[\text{logit } p(\ell = 1 \mid \mathbf{x}, \mathbf{y}_0'', \mathbf{y}_1')]\right)} \right] d\mathbf{x}. \tag{59}$$

For completion, we write the expected win rate improvement for SFT in the same form, connecting the expressions in this section to the result of Theorem 1:

$$\text{Win Rate}_{p(\mathbf{y}_0 \mid \mathbf{x})}[p_{\text{SFT}}(\mathbf{y} \mid \mathbf{x})]$$

$$= \int p(\mathbf{x}) \left[ \frac{\mathbb{E}_{p(\mathbf{y}_1 \mid \mathbf{x})}\left[\mathbb{E}_{p(\mathbf{y}_0 \mid \mathbf{x})} p(\ell = 1 \mid \mathbf{x}, \mathbf{y}_0, \mathbf{y}_1) \mathbb{E}_{p(\mathbf{y}_0' \mid \mathbf{x})}[p(\ell = 1 \mid \mathbf{x}, \mathbf{y}_0', \mathbf{y}_1)]\right]}{\mathbb{E}_{p(\mathbf{y}_1' \mid \mathbf{x})} \mathbb{E}_{p(\mathbf{y}_0'' \mid \mathbf{x})}[p(\ell = 1 \mid \mathbf{x}, \mathbf{y}_0'', \mathbf{y}_1')]} \right] d\mathbf{x}. \tag{60}$$

## G    WIN RATE IMPROVEMENT UPPER BOUND FOR DWRO-KL OBJECTIVES

**Corollary 2.** For any DWRO-KL objective with non-zero KL regularization, the win rate improvement expected over the starting model is upper bounded by:

$$\text{Win Rate}_{p(\mathbf{y}_0 \mid \mathbf{x})}[p^*(\mathbf{y} \mid \mathbf{x})] \le \mathbb{E}_{p(\mathbf{x})}\left[ \max_{\mathbf{y} \in \text{supp}(p(\mathbf{y}_0 \mid \mathbf{x}))} \mathbb{E}_{p(\mathbf{y}_0 \mid \mathbf{x})} p(\ell = 1 \mid \mathbf{x}, \mathbf{y}_0, \mathbf{y}_1) \right].$$

*Proof.* We can rewrite Equation (58) as a weighted average over $\mathbf{y}_1 \in \text{supp}(p(\mathbf{y}_1 \mid \mathbf{x}))$, given $p(\mathbf{y}_1 \mid \mathbf{x})$ is a discrete distribution over sequences. Namely, letting $i \in [1, ..., n]$ index each sequence $\mathbf{y}_1 \in \text{supp}(p(\mathbf{y}_1 \mid \mathbf{x}))$, $w_{i\mathbf{x}} = p(\mathbf{y}_1 \mid \mathbf{x})$, and $z_{i\mathbf{x}} = \exp\left(\frac{1}{\beta}\mathbb{E}_{p(\mathbf{y}_0'' \mid \mathbf{x})} h \cdot p(\ell = 1 \mid \mathbf{x}, \mathbf{y}_0'', \mathbf{y}_1')\right)$, we can rewrite Equation (58) as follows:

$$\text{Win Rate}_{p(\mathbf{y}_0 \mid \mathbf{x})}[p_{\text{DWRO-KL}}(\mathbf{y} \mid \mathbf{x})] = \sum_{\mathbf{x}} p(\mathbf{x}) \frac{\sum_i w_{i\mathbf{x}} z_{i\mathbf{x}} \mathbb{E}_{p(\mathbf{y}_0 \mid \mathbf{x})} p(\ell = 1 \mid \mathbf{x}, \mathbf{y}_0, \mathbf{y}_1 = \mathbf{y}_{[i]})}{\sum_i w_{i\mathbf{x}} z_{i\mathbf{x}}} \tag{61}$$

$$\le \sum_{\mathbf{x}} p(\mathbf{x}) \max_i \mathbb{E}_{p(\mathbf{y}_0 \mid \mathbf{x})} p(\ell = 1 \mid \mathbf{x}, \mathbf{y}_0, \mathbf{y}_1 = \mathbf{y}_{[i]}) \tag{62}$$

$$= \mathbb{E}_{p(\mathbf{x})}\left[ \max_{\mathbf{y} \in \text{supp}(p(\mathbf{y}_0 \mid \mathbf{x}))} \mathbb{E}_{p(\mathbf{y}_0 \mid \mathbf{x})} p(\ell = 1 \mid \mathbf{x}, \mathbf{y}_0, \mathbf{y}_1) \right]. \tag{63}$$

Equation (62) follows from the fact that Equation (61) is a weighted average where $\frac{w_{i\mathbf{x}} z_{i\mathbf{x}}}{\sum_i w_{i\mathbf{x}} z_{i\mathbf{x}}}$ sums to 1. Then, a weighted average is bounded by its max value, concluding the proof. $\qquad\square$

## H COMPARING METHODS

In Table 2, we compare various preference learning methods along the axes of target distribution, objective, and the method of estimating the preference classifier. Ideally, we want a method's target solution is able to put more probability mass over the more preferred sequences; its objective directly seeks to approximate that target over some other goal; and its estimate of the preference classifier is as accurate as possible.

Table 2: Comparison of preference learning algorithms along three dimensions of design choices: target distribution, objective, and estimation of the preference classifier. For readability, we substitute the preference classifier distribution $p(\ell = 1 | \mathbf{x}, \mathbf{y}_0, \mathbf{y})$ with $p_{\text{clf}}$.

| | Target (unnormalized) | Objective (per-query) | Preference Classifier |
|---|---|---|---|
| DWRO-KL | $p(\mathbf{y} \mid \mathbf{x}) \exp\left(\frac{1}{\beta} \mathbb{E}_{p(\mathbf{y}_0 \mid \mathbf{x})}[h \cdot p_{\text{clf}}]\right)$ | $\text{KL}(p_\theta(\mathbf{y} \mid \mathbf{x}) \parallel p^*(\mathbf{y} \mid \mathbf{x}))$ | $\hat{p}_{\text{clf}}$ |
| RLHF | $p(\mathbf{y} \mid \mathbf{x}) \exp\left(\frac{1}{\beta} \mathbb{E}_{p(\mathbf{y}_0 \mid \mathbf{x})}[\text{logit } p_{\text{clf}}]\right)$ | $\text{KL}(p_\theta(\mathbf{y} \mid \mathbf{x}) \parallel p^*(\mathbf{y} \mid \mathbf{x}))$ | $\hat{r}(\mathbf{x}, \mathbf{y})$, BT |
| DPO | $p(\mathbf{y} \mid \mathbf{x}) \exp\left(\frac{1}{\beta} \mathbb{E}_{p(\mathbf{y}_0 \mid \mathbf{x})}[\text{logit } p_{\text{clf}}]\right)$ | $\text{KL}(p^*_{\text{clf}} \parallel p_{\theta,\text{clf}})$ | Equation (9), BT |
| SFT | $p(\mathbf{y} \mid \mathbf{x}) \mathbb{E}_{p(\mathbf{y}_0 \mid \mathbf{x})}[p_{\text{clf}}]$ | $\text{KL}(p^*(\mathbf{y} \mid \mathbf{x}) \parallel p_\theta(\mathbf{y} \mid \mathbf{x}))$ | Bypass |

## I JUDGE MODEL

To train the judge model, we finetune a Pythia-2.8b base model using the same pairwise training set used to train all SFT models by simply modifying the prompt to take in $\mathbf{x}, \mathbf{y}_0, \mathbf{y}_1$ into account. The prompt template is as follows:

```
\n\n Human:  + <Instruction> +
\n\n Candidate Response A: + <response a> +
\n\n Candidate Response B: + <response b> +
\n\n The better answer is Candidate Response
```

The model is trained with a language modeling loss on the output, which is either 'A' or 'B'. For each pair of responses $\mathbf{y}_0, \mathbf{y}_1$, the training set includes two rows, one where $\mathbf{y}_0$ is Response A, and one where $\mathbf{y}_1$ is Response A.

The model is trained using RMSProp with a learning rate of 5e-7 and batch size of 64. The model is trained with a maximum sequence length of 512 and a maximum input length of 511. On the evaluation dataset, the model achieves a per-row classification accuracy of 68.8. (Training the same judge model with a sequence length of 1024 achieves the same accuracy, so we choose to stick with 512 for efficiency.)

To obtain the preference probability of a pair of outputs, we run a forward pass through the judge model twice, once with each order of output pairs, and average the results.

To simulate more opinionated preferences in this preference environment, we sharpen the judge preference probabilities with temperature scaling on the logit-transformed probabilities, with $T = 0.2$:

$$p(\ell = 1 \mid \mathbf{x}, \mathbf{y}_0, \mathbf{y}_1) = \sigma\left(\text{logit } \hat{p}_{\text{judge}}(\ell = 1 \mid \mathbf{x}, \mathbf{y}_0, \mathbf{y}_1)/T\right). \tag{64}$$

## J EXPERIMENT DETAILS

**Dataset processing.** For the Open Assistant dataset (Kopf et al., 2023), we keep only the first turn in each conversation and English-only examples, following Yuan et al. (2024). The dataset only has a train and validation split, so we split the original train set into a train and validation set and leave the validation set for testing / evaluation. The dataset includes multiple candidate responses for each input, all ranked, and to match the pairwise preference learning setup, we create a dataset of all possible pairs for each input. For instance, for a given input with three candidate responses (A, B, C), our paired dataset includes all three pairs (AB, BC, AC). For SHP, we keep only pairs where the score ratio is $> 2$, following Rafailov et al. (2024).

Table 3: Comparison of win rates between RLHF (optimized) and DWRO-KL-BT-logit without additional optimization (non-optimized). RLHF's dropping of the constant in the objective decreases variance in the gradient estimates and generally improves results with small enough $\beta$. However, this change does not yield systematic benefits for every setting, suggesting that there is still room for improvement.

| Dataset | | $\beta = 0.001$ | $\beta = 0.01$ | $\beta = 0.1$ | $\beta = 1$ |
|---|---|---|---|---|---|
| HH | non-optimized | 51.79 (1.12) | 54.59 (0.90) | 68.68 (0.80) | **56.88 (0.46)** |
| | optimized | **70.46 (0.87)** | **63.93 (1.24)** | 69.44 (0.90) | 54.60 (0.67) |
| OASST | non-optimized | 60.74 (3.65) | 61.79 (3.64) | **73.11 (1.16)** | 60.41 (2.05) |
| | optimized | 62.05 (3.65) | 63.95 (3.38) | 66.72 (1.16) | 58.45 (1.82) |

**Training the initial model.** For each dataset, we finetune the base Pythia-2.8b models on all outputs, preferred and dispreferred. The resulting finetuned models serve as our initial models for preference learning. To train these models, we utilize a batch size of 64 and learning rate of 5e-7 chosen based on hyperparameter sweep between [1e-8, 5e-8, 1e-7, 5e-7, 1e-6] on OASST. Following Rafailov et al. (2024), we use the RMSProp optimizer with a learning rate warm up of 150 steps and constant learning rate schedule otherwise. We evaluate every 100 steps and choose the best checkpoint based on validation loss.

**SFT and DPO experiments.** SFT and DPO experiments follow the same learning rate training configuration as the initial model.

**RL experiments.** We use the implementation of reward model training and PPO from the TRL library (von Werra et al., 2020). For reward model training, we use a batch size of 64, learning rate of 5e-7 for Pythia2.8b, and checkpoint every 100 steps, matching the SFT and DPO experiments. For PPO, we use a learning rate=1e-6 (obtained through a hyperparameter sweep of [1e-7, 5e-7, 1e-6] on OASST), batch size=128, and PPOConfig defaults for all other hyperparameters. We checkpoint every five steps and choose checkpoint with the best policy loss (namely, ignoring the estimation of the value head).

**Win rate evaluations.** We sample a set of 100 input prompts from the test set of a given dataset (same 100 prompts for all models) and perform win rate evaluation using the oracle judge for the dataset.

## K    EXPLORING OPTIMIZATION BENEFITS OF RLHF

Below, we compare the win rate results when optimizing the RLHF objective versus a non-optimized DWRO-KL-BT-logit objective that keeps the $\mathbb{E}_{p(\mathbf{y}_0 \mid \mathbf{x})} r(\mathbf{x}, \mathbf{y}_0)$ term. Namely, the RLHF objective we optimize is

$$-\mathcal{L}_{\text{RLHF}}(\theta) = \max_\theta \mathbb{E}_{p(\mathbf{x})}\big[\mathbb{E}_{p_\theta(\mathbf{y} \mid \mathbf{x})}[r(\mathbf{x}, \mathbf{y})] - \beta \text{KL}(p_\theta(\mathbf{y} \mid \mathbf{x}) \parallel p_{\text{ref}}(\mathbf{y} \mid \mathbf{x}))\big]. \tag{65}$$

The non-optimized DWRO-KL-BT-logit objective we optimize is

$$-\mathcal{L}_{\text{RLHF}}(\theta) = \max_\theta \mathbb{E}_{p(\mathbf{x})}\big[\mathbb{E}_{p_\theta(\mathbf{y} \mid \mathbf{x})}[r(\mathbf{x}, \mathbf{y}) - \mathbb{E}_{p(\mathbf{y}_0 \mid \mathbf{x})} r(\mathbf{x}, \mathbf{y}_0)] - \beta \text{KL}(p_\theta(\mathbf{y} \mid \mathbf{x}) \parallel p_{\text{ref}}(\mathbf{y} \mid \mathbf{x}))\big]. \tag{66}$$

Win rate results can be found in Table 3. All experimental details match that of the main paper.

