# OpenReview forum: "Win Rate is All that Can Matter from Preference Data Alone"
_ICLR.cc/2025/Conference — Submitted to ICLR 2025_

### Official Review · Reviewer_41g3 · 2024-10-28

**Soundness:** 3
**Presentation:** 1
**Contribution:** 2
**Rating:** 5
**Confidence:** 3

**Summary:**

The authors present a framework for analyzing different preference learning algorithms. They begin by defining the sampling distribution for pairwise preference learning and introduce a grounded evaluation function, demonstrating that the only grounded function, under a monotonic condition of function h, is the win rate. Building on this result, they propose an objective to optimize the win rate (DWRO) and explore various objective variants. Furthermore, they classify existing popular methods within the DWRO-KL variant. Despite not directly optimizing the DWRO objective, the authors show that SFT can still improve win rates under certain, intuitive conditions. Finally, they provide extensive results based on a single base model.

**Strengths:**

- The paper introduces a general framework that encompasses cases like RLHF or DPO.
- This intuitive framework offers insights into current BT model-based preference learning methods.
- The authors present ablation studies on various functions h and different parameters beta.
- This framework can motivate further studies on finding improved optimization objectives.

**Weaknesses:**

- The framework does not appear to account for unpaired/ranked preference methods.
- The judge model used seems underperforming, achieving only 68.8 on the evaluation set.
- Diverse models could have been employed for more comprehensive results.
- The framework does not address biases in preference learning, such as length bias.
- The paper focuses solely on the DWRO-KL variant without exploring other possible variants.

**Questions:**

- The judge model Pythia 2.8B model seems weak.
- Why do you fine-tune on dispreferred outputs? Which step is that in the framework?
- Stronger judge models should be evaluated for comparison.
- Could you clarify what the anchor distribution is? It is used interchangeably as p or q, which adds to the confusion. Is it solely based on y_0 as defined in Definition 2?
- In the base case of Definition 2, where l=1, there is no y_1 provided. Could you explain this?
- Why is preference limited to the case where l=1? Why doesn’t it include the case when l=0 with x_0 being preferred?
- What is Property 3 in the proof of Proposition 1?
- Since you're using sequence-level KL divergence for DWRO-KL and token-level for PPO, the results may not be directly comparable. Could you comment on this?
- There are some grammatical issues throughout the paper that make comprehension more difficult.
- How many samples are used for each set in your experiments?
- The Appendix mentions Mistral 7B, but no results for this model are included.
- Given that beta is an important parameter, further studies or experiments focusing on its impact would be valuable.
- While DPO is discussed in the analysis, it is not shown in any empirical results. Why is that?

---

> ### Author Response · Authors · 2024-11-25
> **Response to Reviewer 41g3 (part 1)**
>
> Thank you for your review and acknowledging our “general framework” that can “motivate further studies on finding improved optimization objectives.” We address your specific questions and concerns below:
> 1. [The framework does not appear to account for unpaired/ranked preference methods.]
> - We specifically consider the setting of pairwise comparison data in order to formalize what is possible with this form of data in particular. We believe that the community would benefit from the clarity of considering the distinct information offered by different forms of data / feedback / annotations and the implications for the methods that operate on such data. For instance, methods which use direct positive / negative annotations rather than preference pairs can take advantage of the knowledge that a particular sequence is considered globally bad / good and should not / should be generated; this offers different possibilities for evaluation unavailable in the preference pair case (e.g. probability positive sequences should be maximized). We believe future work that conducts an analogous inquiry to ours for other forms of data would be valuable for guiding the field of alignment / post-training methods.
> 2. [The judge model used seems weak / Stronger judge models should be evaluated for comparison.]
> - We agree that the judge model is not a very accurate representation of the preferences encoded in the open-source datasets, but since we treat the resulting judge as our oracle preference classifier and relabel all data with it, its accuracy should not affect the results of our analysis, which focus on comparing different methods including in the ideal setting of having access to the oracle preferences.
> 3. [Diverse models could have been employed for more comprehensive results.]
> - Unfortunately, we were unable to run the same experiments with additional models at this time (the updated experiments add up to over 80 runs, most of which are RL finetuning runs utilizing two A100s for 48 hours each), but the theoretical results hold regardless of model size.
> 4. [The framework does not address biases in preference learning, such as length bias.]
> - Under the framework, we can say the following about length bias: under perfect optimization, the result of an objective such as RLHF or DPO is simply a function of the starting model and the underlying preference classifier for the environment $p(\ell=1|x, y_0, y_1)$. If neither the starting model nor preference classifier has a systematic preference of longer outputs over shorter ones, the framework predicts that the expected solution of existing objectives should not incur a length bias, meaning that any observed length bias is not a fault of the theoretical ideal implied by the objective but rather a result of estimation error (i.e., not reaching the target distribution). If either the starting model or the preference environment has a preference for longer inputs, it is possible to estimate how the preference propagates to the expected target distribution of a DWRO-KL objective like RLHF or DPO; then, a preference for length beyond what is expected can also be attributed to estimation error.
> 5. [The paper focuses solely on the DWRO-KL variant without exploring other possible variants.]
> - Thanks for the feedback. We have added in the discussion of DWRO that the regularization need not be reverse-KL in particular; the chi-squared divergence is another example that could be optimized in the same way.
> 6. [Why do you fine-tune on dispreferred outputs? Which step is that in the framework?]
> - Good question. The reason we finetune on all outputs is to make our initial model approximate the distribution of the preference data samples, thus simulating the case where we improve the model through preference annotations alone.
> 7. [Could you clarify what the anchor distribution is? It is used interchangeably as p or q, which adds to the confusion. Is it solely based on y_0 as defined in Definition 2?]
> - The anchor distribution is the distribution we compare against when measuring win rate. Often, it is the initial start model, as this is generally the one model available other than the current model being trained, but it theoretically could be any model. We’ve updated the notation to be consistent (i.e., always p(y_0|x)).
>
> (continued in part 2)

---

> > ### Author Response · Authors · 2024-11-25
> > **Response to Reviewer 41g3 (part 2)**
> >
> > (continued from part 1)
> >
> > 8. [In the base case of Definition 2, where l=1, there is no y_1 provided. Could you explain this?]
> > - In Definition 2, we define the model in question as $p(y|x)$ instead of $p(y_1|x)$. Then, we write $p(\ell=1|x, y_0, y)$ to denote that $y \sim p(y|x)$ is the response at index 1, to be compared to $y_0 \sim p(y_0|x)$ at index 0. Hope that clarifies things!
> > 9. [Why is preference limited to the case where l=1? Why doesn’t it include the case when l=0 with x_0 being preferred?]
> > - Good question. We could equivalently written all of the math with $p(y_0|x)$ as our model of interest over $p(y_1|x)$ and used $p(\ell=0|x, y_0, y_1)$ to denote the probability that its response is preferred over the anchor. We stick with $\ell=1$ throughout the paper for notational consistency.
> > 10. [What is Property 3 in the proof of Proposition 1?]
> > - Apologies, Property 3 is a typo. The proof should state Property 1 and 2 instead Property 2 and 3. We have fixed this.
> > 11. [Since you're using sequence-level KL divergence for DWRO-KL and token-level for PPO, the results may not be directly comparable. Could you comment on this?]
> > - Good question. Our experiments actually use sequence-level KL divergences so there is no discrepancy between the theoretical and empirical analysis in that regard; the comment about token-level KL divergences in the Appendix referred to an earlier version of the experiments and should have been deleted. We have removed it in the updated paper.
> > 12. [There are some grammatical issues throughout the paper that make comprehension more difficult.]
> > - Thanks for pointing those out. We have gone through to fix many of those.
> > 13. [How many samples are used for each set in your experiments?]
> > - We compute win rate with 100 samples.
> > 14. [The Appendix mentions Mistral 7B, but no results for this model are included.]
> > - We have removed these mentions.
> > 15. [Given that beta is an important parameter, further studies or experiments focusing on its impact would be valuable.]
> > - We have added additional results with more values of $\beta$.
> > 16. [While DPO is discussed in the analysis, it is not shown in any empirical results. Why is that?]
> > - We have additionally added DPO results as well (Figure 1 in the updated paper). We find that despite the fact that RLHF (a DWRO objective) is theoretically preferred over DPO (not a DWRO objective), the results show that RLHF substantially underperforms practically. This leads to the insight that improvements in optimization of DWRO objectives would be especially useful to advance preference learning.
> >
> > Thank you again for your thorough review and great questions. Please let us know if you have any remaining questions or concerns. Otherwise, we would greatly appreciate it if you would consider raising your score.

---

> > > ### Comment · Reviewer_41g3 · 2024-11-27
> > > **Thank you for your answers!**
> > >
> > > I would like to thank the authors for your answers and updates!
> > >
> > > I appreciate answering all the questions! The updated version looks better!
> > >
> > > However, the weaknesses in the review are still part of the given framework, especially 1,2, and 3 (only a single model was evaluated).
> > >
> > > Therefore, I will remain my score.

---

### Official Review · Reviewer_hqd1 · 2024-10-30

**Soundness:** 4
**Presentation:** 3
**Contribution:** 3
**Rating:** 6
**Confidence:** 3

**Summary:**

The authors propose an overarching conceptual framework—a “North Star”—to guide work on learning from human preferences, centered around optimizing for win-rate. This framework consists of a family of Direct Win Rate Optimization (DWRO) objectives for optimizing generative models to achieve higher win-rates. DWRO objectives have of three central components: (1) the strictly monotonic increasing function h, which takes in a preference probability as an input (2) the preference classifier and (3) the anchor distribution over which win-rate is computed.

The authors then present the KL-regularized DWRO (DWRO-KL) objective, which helps mitigates issues related to reward hacking, and which RLHF falls under. Notably the authors assert that optimizing for the DWRO-KL objective is equivalent to approximating a target distribution, which draws an important parallel to variational inference. The authors characterize the target distribution as being the original distribution that has been tilted to have more probability mass placed over sequences that are more likely to be preferred over the anchor distribution. In particular, the notion of sharpness as controlled by the parameter beta and the function h, is used to characterize the target distribution. Under this framing, the authors then introduce the following insights:
1. RLHF is a DWRO-KL objective
2. DPO is a DWRO-KL objective with the same target distribution as RLHF, but used a different DWRO-KL objective than RLHF
3. SFT is not DWRO objective but still optimizes for win-rate via targeting a different distribution

The authors present some empirical results illustrating the effects of varying the DWRO-KL objective on two popular preference learning datasets (i.e., HH and OASST). The authors also provide closed-form solutions for the expected win-rate improvement for DWRO-KL objectives and the SFT objective.

**Strengths:**

The authors present a well-structured, sound, and clear framework for understanding different approaches to learning from preferences. By viewing these methods as either optimizing for win-rate or approximating a specific target distribution, they provide a useful lens for interpreting past and future work in this domain. I appreciate this framework as an original and valuable tool for conceptualizing advancements in preference-based learning.

The authors also establish a concrete parallel between learning from preferences (specifically through optimizing for the DWRO-KL objective) and black-box variational inference. This connection could be especially meaningful for researchers in black-box variational inference, potentially inspiring new research directions. To my knowledge, this connection is novel and holds promise for further impact.

Lastly, the authors offer both closed-form solutions for the expected gain in win-rate for DWRO-KL objectives and empirical results for various DWRO-KL variants. While I am uncertain about the practical utility of these results for other researchers or practitioners, they do provide insights into various design decisions for learning from preferences revolving around the sharpness of the target distribution.

**Weaknesses:**

The paper feels lacking in actionable insights. While the authors offer a useful framework for understanding different approaches to learning from preferences, leading to a range of alternative objectives beyond the standard RLHF objective, RLHF still appears to be the top-performing method. This raises questions about the practical value of the additional flexibility provided by the proposed framework.

Additionally, although the finding that SFT improves win-rate is intriguing, I would appreciate more clarity on why this insight is valuable for the community.

Furthermore, Figure 1 could be clearer. This figure would be significantly improved with guidance on how to interpret it and by highlighting any specific conclusions drawn from it. I was surprised that Figure 1 was not discussed in the results section; perhaps added explanation there would be helpful.

**Questions:**

What practical insights can we draw from the provided closed-form solutions for win-rate improvement for the DWRO-KL objectives?

---

> ### Author Response · Authors · 2024-11-25
> **Response to Reviewer hqd1**
>
> Thank you for your review and for appreciating the proposed framework as a “useful lens for interpreting past and future work in this domain.” And thank you for your insightful questions and comments; they have definitely helped us strengthen the paper. We address each below:
> 1. [The paper feels lacking in actionable insights. While the authors offer a useful framework for understanding different approaches to learning from preferences, leading to a range of alternative objectives beyond the standard RLHF objective, RLHF still appears to be the top-performing method. This raises questions about the practical value of the additional flexibility provided by the proposed framework.]
> - Thanks for the feedback. We’ve incorporated the following three actionable insights to the paper discussion:
>   - The (expanded and updated) comparisons of DWRO objectives show that optimization success is more indicative of performance than the spectrum of design choices around DWRO, suggesting that perhaps the highest leverage strategies for improving DWRO methods (including RLHF) is improving optimization rather than proposing a new variant of such an objective.
>   - The fact that RLHF > DPO > SFT based on closeness to DWRO but SFT > DPO > RLHF based on ease of optimization points to the potential benefit of mixing and matching methods, a strategy that is already used (e.g. SFT then RLHF, DPO + SFT) but could possibly be leveraged further.
>   - Overall, the framework suggests the following directions for future work: 1. Improving the optimization of an objective that already directly optimizes for win rate, or 2. developing easier-to-optimize objectives that are closer to direct win optimization than existing alternatives.
> 2. [Additionally, although the finding that SFT improves win-rate is intriguing, I would appreciate more clarity on why this insight is valuable for the community.]
> - Thanks for the feedback. The primary insight comes from the analytical expression for the win rate improvement itself. We’ve added the following implications to the discussion following this theorem:
>   - The win rate improvement from preference annotations alone is a function of the starting model.
>   - This improvement is bounded for SFT based on properties of this starting model. In other words, SFT on a model’s preferred samples has limits in terms of win rate improvement, no matter how many annotations of preference pairs are collected. This is also true for RLHF or DPO with a fixed, non-zero beta.
>   - The result characterizes a particular form of diversity (namely, in preference) that is important for self-improvement. On one extreme, if a model only ever outputs responses that are equally preferred to each other, no improvement is possible. The more diverse in preference the starting model, the more improvement possible.
> 3. [Furthermore, Figure 1 could be clearer. This figure would be significantly improved with guidance on how to interpret it and by highlighting any specific conclusions drawn from it. I was surprised that Figure 1 was not discussed in the results section; perhaps added explanation there would be helpful.]
> - Good point. Figure 1 is meant to show in a toy example how the target distributions of different objectives vary. We’ve added more discussion around the figure, but to make room for other more important text (e.g., more sign-posting around takeaways), we’ve decided to move it and the relevant discussion to the appendix.
> 4. [What practical insights can we draw from the provided closed-form solutions for win-rate improvement for the DWRO-KL objectives?]
> - Thanks for the question. The primary insight is that the introduction of the KL regularization means that the win rate possible must be a function of the support of the starting model; we’ve formalized this as Corollary 1 in the updated paper. Concretely, the best these objectives can do theoretically is defined by how well the best response under the original model can do.
>
> Thanks again for your feedback and questions. If you believe we have sufficiently addressed them, we would really appreciate it if you considered raising your score.

---

> > ### Comment · Reviewer_hqd1 · 2024-11-27
> >
> > Thank you for your response! I am keeping my score as is.
> >
> > > The paper feels lacking in actionable insights
> >
> > I appreciate your added analysis, but my concern here still holds as the main justification for the score I gave; The insight that "the highest leverage strategies for improving DWRO methods (including RLHF) is improving optimization" is interesting, but the RLHF community has already been focused primarily on improving optimization instead of some other DWRO design choice so I do not see this as a particularly actionable insight.
> >
> > In general, this paper provides a nice---although as other reviewers have pointed out not entirely novel---theoretical framework for understanding approaches to learning from human feedback (hence a score of 6). But I do not see any key takeaways from this paper that I think will provide new insights/directions to researchers in the RLHF community, including myself (hence not higher than 6).

---

### Official Review · Reviewer_QHAU · 2024-10-31

**Soundness:** 1
**Presentation:** 1
**Contribution:** 1
**Rating:** 1
**Confidence:** 4

**Summary:**

This paper studies preference learning algorithms for language models. The authors propose a family of objectives called DWRO to directly optimize the win rate. The authors discuss the relationship between DWRO, reinforcement learning from human feedback (RLHF), and supervised fine-tuning (SFT), examining how SFT can improve win rates. They also conduct experiments to evaluate different DWRO variants.

**Strengths:**

a. This paper studies a general preference learning problem, with the goal of improving win rates. The widely-used Bradley-Terry model is a special case within this general preference framework.

b. The authors conduct experiments to evaluate various DWRO variants with different choices of $\psi$.

**Weaknesses:**

a. The major contribution, DWRO, is not new, as it has already been proposed (see Equation 6, the preference optimization objective, in [1]). The function $\Psi$ serves the same role as $h$. Additionally, the relationship between the objective and RLHF has also been discussed in [1].

b. Section 3.3, which discusses SFT, is unclear. In line 313, the authors state that “SFT is analogous to estimating the distribution $p(\mathbf{y}_1 \mid \mathbf{x}, \ell=1)$” but do not provide justification. The quantity $p(\mathbf{y}_1 \mid \mathbf{x}, \ell=1)$ is confusing, as it’s unclear how another response, $y_0$, is sampled. Is it sampled from a fixed reference distribution? More details are needed on how SFT is performed on preferred samples. Furthermore, the justification for why Eq. 16 is the target distribution for SFT is lacking. The authors show equivalence between Eq. 15 and Eq. 16 but do not explain the derivation of Eq. 16.

c. Experimental details are also insufficient. There is no explanation of how the reference model is trained—does it use the same preference dataset as the models with DWRO variants? In addition, details on constructing the preference pairs for training are missing. The purpose of the experiments is also unclear. If the goal is to compare different DWRO objectives, it would be more meaningful to have them compete directly with each other, rather than against a fixed reference model. Furthermore, the judge model, Pythia-2.8b, is lightweight, whereas higher-standard models like GPT-4 are typically used as judges for evaluating win rates.

d. The overall presentation requires improvement. Section 2.2 all talks about the evaluation function $\phi$, but it does not appear in later sections. Additionally, the notation for the anchor distribution is inconsistent, with $q$ used in some expressions and $p$ in others, which makes the paper difficult to follow.

[1] Azar, Mohammad Gheshlaghi, et al. "A general theoretical paradigm to understand learning from human preferences." International Conference on Artificial Intelligence and Statistics. PMLR, 2024.

**Questions:**

See weaknesses.

---

> ### Author Response · Authors · 2024-11-25
> **Response to Reviewer QHAU**
>
> Thank you for your review. We wish to highlight that our paper is not a methods paper but rather an analysis paper. We have updated the text to further emphasize that point, and we hope you will reconsider your review in light of this. Responding to your individual comments below:
> 1. [The major contribution, DWRO, is not new, as it has already been proposed (see Equation 6, the preference optimization objective, in [1]). The function Ψ serves the same role as h. Additionally, the relationship between the objective and RLHF has also been discussed in [1].]
> - DWRO is not the major contribution of this paper, nor is its relationship to RLHF. As mentioned in the related work, we agree with the reviewer that DWRO-KL is equivalent to $\Psi$-PO from [1]. We also cite [1] for the derivation of the relationship between RLHF and DWRO-KL in the RLHF analysis section. Instead, the primary contribution of this work is to provide a mathematical justification for win rate as the central object in preference learning and re-characterize existing methods from this vantage point.
> 2. [Section 3.3, which discusses SFT, is unclear. In line 313, the authors state that “SFT is analogous to estimating the distribution $p(y_1∣x, \ell=1)$ but do not provide justification. The quantity $p(y_1∣x, \ell=1)$ is confusing, as it’s unclear how another response, $y_0$, is sampled. Is it sampled from a fixed reference distribution? More details are needed on how SFT is performed on preferred samples. Furthermore, the justification for why Eq. 16 is the target distribution for SFT is lacking. The authors show equivalence between Eq. 15 and Eq. 16 but do not explain the derivation of Eq. 16.]
> - Given a joint distribution over $x, y_0, y_1, \ell$ where $\ell=1$ denotes that $y_1$ is preferred, the distribution over the preferred sample in a pair is by definition the conditional distribution $p(y_1|x, \ell=1)$. $y_0$ is sampled from the distribution $p(y_0|x)$ that is part of the joint sampling distribution of $x, y_0, y_1, \ell$. Performing SFT on the preferred sample in a pair targets $p(y_1|x, \ell=1)$ via maximum likelihood estimation. Eq 16 (now Eq 12) follows directly from the RHS of the numerator in Eq 14 (now Eq 13).
> 3. [Experimental details are also insufficient. There is no explanation of how the reference model is trained—does it use the same preference dataset as the models with DWRO variants?]
> - The reference model is the starting point for all the DWRO runs. Thus, the win rate is computed between the model before and after DWRO.
> 4. [In addition, details on constructing the preference pairs for training are missing.]
> - Appendix G describes how the datasets are processed to be preference pairs.
> 5. [The purpose of the experiments is also unclear. If the goal is to compare different DWRO objectives, it would be more meaningful to have them compete directly with each other, rather than against a fixed reference model.]
> - We are interested in evaluating the methods based on the objective they seek to approximate. In this case, that objective is maximizing win rate over their starting point, which is why we evaluate the methods this way.
> 6. [Furthermore, the judge model, Pythia-2.8b, is lightweight, whereas higher-standard models like GPT-4 are typically used as judges for evaluating win rates.]
> - Our analysis is applicable for any preference environment defined by any arbitrary preference classifier $p(\ell = 1 | x, y_0 , y_1)$. We agree that GPT-4 is typically a better judge for approximating the average human preference, but it is only a proxy. We train our own judge model and make it the oracle preference so that we have easy access to the truth for analysis. As mentioned in section 4.1, “To simulate a preference environment, we train an oracle judge model per dataset to estimate $p(\ell = 1 | x, y_0 , y_1)$ and relabel the preference annotations in the dataset using this judge model.”
> 7. [The overall presentation requires improvement. Section 2.2 all talks about the evaluation function
> $\phi$, but it does not appear in later sections. Additionally, the notation for the anchor distribution is inconsistent, with q used in some expressions and p in others, which makes the paper difficult to follow.]
> - We do not use $\phi$ in later sections because we have no need to discuss an evaluation function in general once we’ve determined that the evaluation function must be win rate. We have updated the notation for the anchor distribution to be consistent–thanks for the feedback!
>
> Thank you for your comments, which helped us realize that there were parts of the paper we could clarify to avoid the possibility of confusion. We hope that, after reading this response and looking over the updated paper, you will reconsider your review.

---

> > ### Comment · Reviewer_QHAU · 2024-11-27
> >
> > Appendix G is entirely focused on proofs, and I don’t think it contains details about constructing preference pairs.
> >
> > Regarding the judge model, my concern is that it is only a 2.8B model, and I am worried about whether it is powerful enough to provide accurate judgments.
> >
> > The proposed win-rate-centric framework is not new; it is essentially general preference learning, as proposed by Azar et al., 2024. Similarly, the DWRO objective is exactly the same as the IPO objective. Section 5 mainly states that DPO and SFT do not always improve win rates, but this is very expected since they are not designed for general preference learning.
> >
> > Section 6 evaluates DPO, SFT, and other variants of IPO objectives. However, the goal of the IPO objective is to calculate the best response to a reference model. In contrast, there is a line of work focused on learning the Nash equilibrium, which is much more general and powerful than the best response to a fixed policy.
> >
> > Overall, the contribution of this paper is very limited, given that general preference learning has been widely studied and this paper still focuses on studying variations of the initial IPO objective.

---

### Official Review · Reviewer_WXDN · 2024-11-04

**Soundness:** 2
**Presentation:** 1
**Contribution:** 1
**Rating:** 3
**Confidence:** 5

**Summary:**

This paper presents a framework for preference learning centered around the concept of "win rate" as the fundamental metric derived from preference data. Recognizing that preference learning has become increasingly complex, the authors propose Direct Win Rate Optimization (DWRO) as a more straightforward family of objectives for preference learning. They demonstrate that Reinforcement Learning from Human Feedback (RLHF) is a KL-regularized DWRO objective, unlike Supervised Fine-Tuning (SFT), although SFT still improves win rate. Through theoretical analysis, the paper explores how various preference learning objectives impact the sharpness of target distributions and provides closed-form solutions for expected win rate improvements. The authors support their framework with empirical results, showing that alternative KL-regularized DWRO objectives can outperform RLHF, and conclude by offering guidance for future research based on their win-rate-centric perspective.

**Strengths:**

1. Theoretical Contributions: It offers formal proofs showing that only the win rate is grounded in preference data, providing justification for the proposed DWRO framework.

2. Empirical Validation: The experimental results support the theoretical claims and illustrate the efficacy of DWRO objectives compared to RLHF.

**Weaknesses:**

1. The writing is poor. Especially for the contribution is poor, first, the authors should use mathematical formulas to accurately show their novelty instead of confusing words; second, the contribution should be split into fewer points, like 3 to 4. Now 7 points are too many.

2. The novelty of this work is little. Although the authors claim that they are the first to consider the win rate, the win rate is basically the same as the general preference, which is already been studied by a series work of Nash learning, such as Munos 484 et al. (2023), [1], [2], [3]. In this work, the authors fix the policy of one response as the based one and optimize the others, this is exactly the same as each iteration of Nash learning optimization of Munos 484 et al. (2023). Besides, although the authors consider different functions h, all the types they list in Table 3 have been considered in the previous works.

3. Based on the theory of Munos 484 et al. (2023) and [3], fixing one response and optimizing the other may not be efficient and achieve the best performance since one iteration will not lead to the Nash equilibrium. Hence, the authors are suggested to compare the performance of their method to Munos 484 et al. (2023), and [3].



[1] Ye, Chenlu, et al. "A theoretical analysis of nash learning from human feedback under general kl-regularized preference." arXiv preprint arXiv:2402.07314 (2024).

[2] Rosset, Corby, et al. "Direct nash optimization: Teaching language models to self-improve with general preferences." arXiv preprint arXiv:2404.03715 (2024).

[3] Calandriello D, Guo D, Munos R, et al. Human alignment of large language models through online preference optimisation[J]. arXiv preprint arXiv:2403.08635, 2024.

**Questions:**

1. Line 124: If h is a function, what is its' mapping: from what space to what space?

2. Line 354: the notation for the variance is confusing and I cannot find a clarification. To which variance the authors are taking variance?

3. Some typos: Line 228 should be "between the model and the target distribution";  Line 354 and 357, ".5" is somehow unprofessional and easily leads to confusion. It is better to change it to 0.5 or 1/2.

---

> ### Author Response · Authors · 2024-11-25
> **Response to Reviewer WXDN**
>
> Thank you for your review and for recognizing the formal proof about win rate as a strength of the paper. As we mentioned in the overall response, however, we believe that there may have been some confusion over what the paper claims as its core contributions. We hope our response to each of your points below, as well as our updated paper, will provide more clarity:
> 1. [The writing is poor. Especially for the contribution is poor, first, the authors should use mathematical formulas to accurately show their novelty instead of confusing words; second, the contribution should be split into fewer points, like 3 to 4. Now 7 points are too many.]
> - Thanks for the feedback. We have updated the introduction to include only the primary contributions of the paper, and we have made sure to avoid any imprecise language.
> 2. [The novelty of this work is little. Although the authors claim that they are the first to consider the win rate, the win rate is basically the same as the general preference, which is already been studied by a series work of Nash learning, such as Munos 484 et al. (2023), [1], [2], [3].]
> - First, we do not claim that we are the first to consider win rate. As mentioned in the related work: “While win rate is already a central evaluation in preference learning (Li et al., 2023; Zheng et al., 2024), in our work we underscore that it is the only evaluation grounded in the sampling distribution itself.” Moreover, we already cite Munos et al in the related work and have added the other contemporary works as well. More fundamentally, however, our paper is not a methods paper; instead, the novelty of our work lies in our win-rate centric analysis and its resulting takeaways.
> 3. [Based on the theory of Munos 484 et al. (2023) and [3], fixing one response and optimizing the other may not be efficient and achieve the best performance since one iteration will not lead to the Nash equilibrium. Hence, the authors are suggested to compare the performance of their method to Munos 484 et al. (2023), and [3].]
> - The point of our paper is not to propose a new method, but rather to understand the preference learning landscape through the lens of directly optimizing for win rate. We do not suggest that one should choose one variant of direct win rate optimization over another–in fact, our experiments (updated to consider additional settings) show that the particular variant matters less than the success of optimization for a given run. The insight that preference learning could benefit from improved strategies for optimization is generally applicable, including for the NLHF method proposed in Munos et al 2023.
> 4. [Line 124: If h is a function, what is its' mapping: from what space to what space?]
> - $h$ maps from [0, 1] to $\mathbb{R}$.
> 5. [Line 354: the notation for the variance is confusing and I cannot find a clarification. To which variance the authors are taking variance?]
> - The variance is with respect to the distribution $p(y_1|x)$, as denoted in the subscript. This variance term can be understood procedurally as follows: for each response $y_1$ from $p(y_1|x)$, we compute its average preference over $p(y_0|x)$ by computing preference probabilities for each pair and averaging by the prevalence of $y_0$ under the initial model. Then, we take the variance across all $y_1$s of these average preferences. In short, it is the variance in average preference across responses.
> 6. [Some typos]
> - Thanks, fixed!
>
> We appreciate your feedback, which highlighted parts of the paper that we could clarify. In light of our response and paper update, we hope you will consider raising your score.

---

> > ### Comment · Reviewer_WXDN · 2024-11-26
> >
> > Thank the authors for their reply. Since the win rate method and the intuitions have been studied by a line of work, I remain my score.

---

### Author Response · Authors · 2024-11-25
**General response**

Thank you all for your reviews. We are especially pleased that 41g328 acknowledges that the paper presents a “general” and “intuitive” framework and hqd1 appreciates our proposed framework as “an original and valuable tool for conceptualizing advancements in preference-based learning.” We want to emphasize that the **proposed win-rate centric framework is the main contribution of our work; this paper is not a methods paper.** To summarize and clarify our primary contributions:
1. We are the first to point out that given simple desiderata, win rate is the only evaluation that is mathematically grounded in the preference data sampling distribution alone. Any other evaluation must be a function of external choices or assumptions.
- We do not claim to be the first to discuss win rate as an evaluation or an objective, but we are the first to present its unique position for preference learning. This result provides a focal point to understand this otherwise seemingly elaborate field.
2. We present a win-rate centric analysis of common preference learning algorithms, with insights that are broadly relevant to variants of these algorithms as well. Our analysis proceeds by introducing direct win rate optimization (DWRO) as the theoretical ideal for preference learning and describing how common algorithms relate to or deviate from it, as well as what this implies about their benefits and limitations.
- As correctly pointed out by WXDN and QHAU, as well as mentioned in our related work, other works that have proposed directly optimizing win rate. We do not claim DWRO as a methodological contribution but rather as an analytical tool. We are the first to our knowledge to motivate and use DWRO as the starting point to understand the preference learning landscape.

Our analysis offers a range of results and takeaways:
1. **There are theoretical benefits to being as close to DWRO as possible.** These include (1) optimizing train loss corresponds to optimizing for the test evaluation we care about (up to noise and overfitting); and (2) its solution has no theoretical limits as regularization strength goes to zero.
- RLHF is a variant of DWRO under assumptions, meaning it enjoys the above benefits. However, there are other variants that could work too.
- DPO is not a DWRO; this provides insight into why DPO loss is bad for checkpointing, i.e., (1) does not hold.
- SFT is not a DWRO either; while it can improve win rate, there are limits.
2. **We can characterize the expected win rate improvement of common preference learning algorithms like RLHF, DPO, and SFT.** This result highlights that **improvement depends on the initial model.** In the case of SFT in particular, win rate improvement is exactly a function of the variance of average preference probabilities, providing a formal definition of the role of diversity in self-improvement.
3. Preference learning objectives (e.g., KL-regularized DWRO) can be mapped to those in probabilistic inference (e.g., variational inference). This presents an opportunity to use insights from the latter to improve the former.

In response to reviewer comments, we made the following additions / modifications:
1. Previously, the introduction provided a step-by-step summary of our analysis, which we viewed as novel in itself. We have now modified the introduction to focus on only the primary contributions and takeaways of the paper.
2. We have moved the related work section to the beginning of the paper, to emphasize the differences with prior work before the analysis begins.
3. In each section of the analysis, we have added a “Takeaways” paragraph to make it easier to recognize the contributions of the analysis.
4. We have expanded the experimental section. We:
- Increased the number of DWRO-KL settings tested (more values of beta, oracle and non-oracle preference classifier)
- Added comparisons to DPO and SFT

With these additional experimental results, we have added a new empirically-driven contribution to the work: **optimization success plays a primary role in the success of preference learning algorithms.** DWRO objectives underperform empirically relative to theoretical expectations, and across DWRO variants (including RLHF), the train loss achieved by a given run is more predictive of test win rate than any setting of regularization, regularization strength, preference classifier estimate, or $h$ transformation. This suggests that although we may have a theoretical ideal for learning from preference data, much work remains to fully capitalize on these objectives in the form of optimization improvements. Moreover, the fact that RLHF > DPO > SFT in their relation to DWRO but SFT > DPO > RLHF in ease of optimization gives a useful perspective on the benefit of mixing and matching methods (e.g., SFT then RLHF, DPO + SFT).

We look forward to engaging in discussion and would deeply appreciate it if the reviewers would reconsider their reviews / raise their scores with the above in mind.

---

### Meta-Review · Area_Chair_akzt · 2024-12-16

**Metareview:**

This paper proposes a general framework for preference learning with the win rate. The authors instantiate several existing algorithms in their framework, analyze them, and support their findings empirically. The scores of this paper are 6, 5, 3, and 1; and did not change during the discussion. The reviewers had several major concerns:

* **Poor writing:** Notation is sometimes inconsistent and not all technical novelty is stated mathematically.

* **Novelty:** Many insights in this work already appeared in papers since 2023.

* **Experiments:** Not all experimental details are stated. The judge (Pythia-2.8b) is weak.

I agree with the authors that there is novelty in their work over prior works. However, it needs to be clearly stated and well evaluated. This is not the case now and therefore the paper cannot be accepted.

**Additional Comments On Reviewer Discussion:**

See the meta-review for details.

---

### Decision · Program_Chairs · 2025-01-22

Reject